# Algorithmic Regularization in Learning Deep Homogeneous Models: Layers are Automatically Balanced*

**Simon S. Du**[†]  **Wei Hu**[‡]  **Jason D. Lee**[§]

## Abstract

We study the implicit regularization imposed by gradient descent for learning multi-layer homogeneous functions including feed-forward fully connected and convolutional deep neural networks with linear, ReLU or Leaky ReLU activation. We rigorously prove that gradient flow (i.e. gradient descent with infinitesimal step size) effectively enforces the differences between squared norms across different layers to remain *invariant* without any explicit regularization. This result implies that if the weights are initially small, gradient flow automatically balances the magnitudes of all layers. Using a discretization argument, we analyze gradient descent with positive step size for the non-convex low-rank asymmetric matrix factorization problem without any regularization. Inspired by our findings for gradient flow, we prove that gradient descent with step sizes $\eta_t = O\left(t^{-\left(\frac{1}{2}+\delta\right)}\right)$ $(0 < \delta \leq \frac{1}{2})$ automatically balances two low-rank factors and converges to a bounded global optimum. Furthermore, for rank-1 asymmetric matrix factorization we give a finer analysis showing gradient descent with constant step size converges to the global minimum at a globally linear rate. We believe that the idea of examining the invariance imposed by first order algorithms in learning homogeneous models could serve as a fundamental building block for studying optimization for learning deep models.

## 1 Introduction

Modern machine learning models often consist of multiple layers. For example, consider a feed-forward deep neural network that defines a prediction function

$$\boldsymbol{x} \mapsto f(\boldsymbol{x}; \boldsymbol{W}^{(1)}, \ldots, \boldsymbol{W}^{(N)}) = \boldsymbol{W}^{(N)}\phi(\boldsymbol{W}^{(N-1)} \cdots \boldsymbol{W}^{(2)}\phi(\boldsymbol{W}^{(1)}\boldsymbol{x}) \cdots),$$

where $\boldsymbol{W}^{(1)}, \ldots, \boldsymbol{W}^{(N)}$ are weight matrices in $N$ layers, and $\phi(\cdot)$ is a point-wise *homogeneous* activation function such as Rectified Linear Unit (ReLU) $\phi(x) = \max\{x, 0\}$. A simple observation is that this model is *homogeneous*: if we multiply a layer by a positive scalar $c$ and divide another layer by $c$, the prediction function remains the same, e.g. $f(\boldsymbol{x}; c\boldsymbol{W}^{(1)}, \ldots, \frac{1}{c}\boldsymbol{W}^{(N)}) = f(\boldsymbol{x}; \boldsymbol{W}^{(1)}, \ldots, \boldsymbol{W}^{(N)})$.

A direct consequence of homogeneity is that a solution can produce small function value while being unbounded, because one can always multiply one layer by a huge number and divide another

[†]Machine Learning Department, School of Computer Science, Carnegie Mellon University. Email: ssdu@cs.cmu.edu

[‡]Computer Science Department, Princeton University. Email: huwei@cs.princeton.edu

[§]Department of Data Sciences and Operations, Marshall School of Business, University of Southern California. Email: jasonlee@marshall.usc.edu

layer by that number. Theoretically, this possible unbalancedness poses significant difficulty in analyzing first order optimization methods like gradient descent/stochastic gradient descent (GD/SGD), because when parameters are not a priori constrained to a compact set via either coerciveness[5] of the loss or an explicit constraint, GD and SGD are not even guaranteed to converge [Lee et al., 2016, Proposition 4.11]. In the context of deep learning, Shamir [2018] determined that the primary barrier to providing algorithmic results is in that the sequence of parameter iterates is possibly unbounded.

Now we take a closer look at asymmetric matrix factorization, which is a simple two-layer homogeneous model. Consider the following formulation for factorizing a low-rank matrix:

$$\min_{\boldsymbol{U}\in\mathbb{R}^{d_1\times r}, \boldsymbol{V}\in\mathbb{R}^{d_2\times r}} f\left(\boldsymbol{U},\boldsymbol{V}\right) = \frac{1}{2}\left\|\boldsymbol{U}\boldsymbol{V}^\top - \boldsymbol{M}^*\right\|_F^2, \tag{1}$$

where $\boldsymbol{M}^* \in \mathbb{R}^{d_1\times d_2}$ is a matrix we want to factorize. We observe that due to the homogeneity of $f$, it is not smooth[6] even in the neighborhood of a globally optimum point. To see this, we compute the gradient of $f$:

$$\frac{\partial f\left(\boldsymbol{U},\boldsymbol{V}\right)}{\partial \boldsymbol{U}} = \left(\boldsymbol{U}\boldsymbol{V}^\top - \boldsymbol{M}^*\right)\boldsymbol{V}, \qquad \frac{\partial f\left(\boldsymbol{U},\boldsymbol{V}\right)}{\partial \boldsymbol{V}} = \left(\boldsymbol{U}\boldsymbol{V}^\top - \boldsymbol{M}^*\right)^\top \boldsymbol{U}. \tag{2}$$

Notice that the gradient of $f$ is not homogeneous anymore. Further, consider a globally optimal solution $(\boldsymbol{U}, \boldsymbol{V})$ such that $\|\boldsymbol{U}\|_F$ is of order $\epsilon$ and $\|\boldsymbol{V}\|_F$ is of order $1/\epsilon$ ($\epsilon$ being very small). A small perturbation on $\boldsymbol{U}$ can lead to dramatic change to the gradient of $\boldsymbol{U}$. This phenomenon can happen for all homogeneous functions when the layers are unbalanced. The lack of nice geometric properties of homogeneous functions due to unbalancedness makes first-order optimization methods difficult to analyze.

A common theoretical workaround is to artificially modify the natural objective function as in (1) in order to prove convergence. In [Tu et al., 2015, Ge et al., 2017a], a regularization term for balancing the two layers is added to (1):

$$\min_{\boldsymbol{U}\in\mathbb{R}^{d_1\times r}, \boldsymbol{V}\in\mathbb{R}^{d_2\times r}} \frac{1}{2}\left\|\boldsymbol{U}\boldsymbol{V}^\top - \boldsymbol{M}\right\|_F^2 + \frac{1}{8}\left\|\boldsymbol{U}^\top\boldsymbol{U} - \boldsymbol{V}^\top\boldsymbol{V}\right\|_F^2. \tag{3}$$

For problem (3), the regularizer removes the homogeneity issue and the optimal solution becomes unique (up to rotation). Ge et al. [2017a] showed that the modified objective (3) satisfies (i) every local minimum is a global minimum, (ii) all saddle points are strict[7], and (iii) the objective is smooth. These imply that (noisy) GD finds a global minimum [Ge et al., 2015, Lee et al., 2016, Panageas and Piliouras, 2016].

On the other hand, empirically, removing the homogeneity is not necessary. We use GD with random initialization to solve the optimization problem (1). Figure 1a shows that even *without* regularization term like in the modified objective (3) GD with random initialization converges to a global minimum and the convergence rate is also competitive. A more interesting phenomenon is shown in Figure 1b in which we track the Frobenius norms of $\boldsymbol{U}$ and $\boldsymbol{V}$ in all iterations. The plot shows that the ratio between norms remains a constant in all iterations. Thus the unbalancedness does not occur at all! In many practical applications, many models also admit the homogeneous property (like deep neural networks) and first order methods often converge to a balanced solution. A natural question arises:

**Why does GD balance multiple layers and converge in learning homogeneous functions?**

In this paper, we take an important step towards answering this question. Our key finding is that the gradient descent algorithm provides an implicit regularization on the target homogeneous function. First, we show that on the gradient flow (gradient descent with infinitesimal step size) trajectory induced by any differentiable loss function, for a large class of homogeneous models, including fully connected and convolutional neural networks with linear, ReLU and Leaky ReLU activations, the differences between squared norms across layers remain invariant. Thus, as long as at the beginning the differences are small, they remain small at all time. Note that small differences arise in commonly used initialization schemes such as $\frac{1}{\sqrt{d}}$ Gaussian initialization or

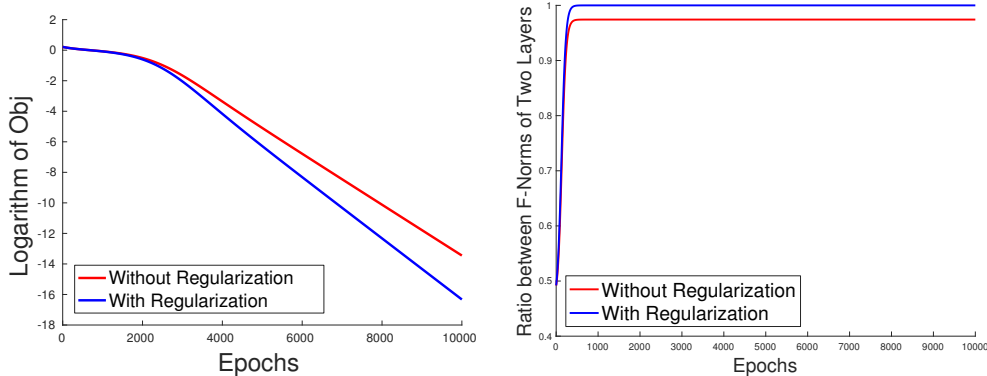

(a) Comparison of convergence rates of GD for objective functions (1) and (3).

(b) Comparison of quantity $\|\boldsymbol{U}\|_F^2 / \|\boldsymbol{V}\|_F^2$ when running GD for objective functions (1) and (3).

Figure 1: Experiments on the matrix factorization problem with objective functions (1) and (3). Red lines correspond to running GD on the objective function (1), and blue lines correspond to running GD on the objective function (3).

Xavier/Kaiming initialization schemes [Glorot and Bengio, 2010, He et al., 2016]. Our result thus explains why using ReLU activation is a better choice than sigmoid from the optimization point view. For linear activation, we prove an even stronger invariance for gradient flow: we show that $\boldsymbol{W}^{(h)}(\boldsymbol{W}^{(h)})^\top - (\boldsymbol{W}^{(h+1)})^\top \boldsymbol{W}^{(h+1)}$ stays invariant over time, where $\boldsymbol{W}^{(h)}$ and $\boldsymbol{W}^{(h+1)}$ are weight matrices in consecutive layers with linear activation in between.

Next, we go beyond gradient flow and consider gradient descent with positive step size. We focus on the asymmetric matrix factorization problem (1). Our invariance result for linear activation indicates that $\boldsymbol{U}^\top \boldsymbol{U} - \boldsymbol{V}^\top \boldsymbol{V}$ stays unchanged for gradient flow. For gradient descent, $\boldsymbol{U}^\top \boldsymbol{U} - \boldsymbol{V}^\top \boldsymbol{V}$ can change over iterations. Nevertheless we show that if the step size decreases like $\eta_t = O\left(t^{-\left(\frac{1}{2}+\delta\right)}\right)$ $(0 < \delta \leqslant \frac{1}{2})$, $\boldsymbol{U}^\top \boldsymbol{U} - \boldsymbol{V}^\top \boldsymbol{V}$ will remain small in all iterations. In the set where $\boldsymbol{U}^\top \boldsymbol{U} - \boldsymbol{V}^\top \boldsymbol{V}$ is small, the loss is coercive, and gradient descent thus ensures that all the iterates are bounded. Using these properties, we then show that gradient descent converges to a globally optimal solution. Furthermore, for rank-1 asymmetric matrix factorization, we give a finer analysis and show that randomly initialized gradient descent with *constant* step size converges to the global minimum at a globally linear rate.

**Related work.** The homogeneity issue has been previously discussed by Neyshabur et al. [2015a,b]. The authors proposed a variant of stochastic gradient descent that regularizes paths in a neural network, which is related to the max-norm. The algorithm outperforms gradient descent and AdaGrad on several classification tasks.

A line of research focused on analyzing gradient descent dynamics for (convolutional) neural networks with one or two unknown layers [Tian, 2017, Brutzkus and Globerson, 2017, Du et al., 2017a,b, Zhong et al., 2017, Li and Yuan, 2017, Ma et al., 2017, Brutzkus et al., 2017]. For one unknown layer, there is no homogeneity issue. While for two unknown layers, existing work either requires learning two layers separately [Zhong et al., 2017, Ge et al., 2017b] or uses re-parametrization like weight normalization to remove the homogeneity issue [Du et al., 2017b]. To our knowledge, there is no rigorous analysis for optimizing multi-layer homogeneous functions.

For a general (non-convex) optimization problem, it is known that if the objective function satisfies (i) gradient changes smoothly if the parameters are perturbed, (ii) all saddle points and local maxima are strict (i.e., there exists a direction with negative curvature), and (iii) all local minima are global (no spurious local minimum), then gradient descent [Lee et al., 2016, Panageas and Piliouras, 2016] converges to a global minimum. There have been many studies on the optimization landscapes of neural networks [Kawaguchi, 2016, Choromanska et al., 2015, Du and Lee, 2018, Hardt and Ma, 2016, Bartlett et al., 2018, Haeffele and Vidal, 2015, Freeman and Bruna, 2016, Vidal et al., 2017, Safran and Shamir, 2016, Zhou and Feng, 2017, Nguyen and Hein, 2017a,b, Zhou and Feng, 2017, Safran and Shamir, 2017], showing that the objective functions have properties (ii) and (iii).

Nevertheless, the objective function is in general not smooth as we discussed before. Our paper complements these results by showing that the magnitudes of all layers are balanced and in many cases, this implies smoothness.

**Paper organization.** The rest of the paper is organized as follows. In Section 2, we present our main theoretical result on the implicit regularization property of gradient flow for optimizing neural networks. In Section 3, we analyze the dynamics of randomly initialized gradient descent for asymmetric matrix factorization problem with unregularized objective function (1). In Section 4, we empirically verify the theoretical result in Section 2. We conclude and list future directions in Section 5. Some technical proofs are deferred to the appendix.

**Notation.** We use bold-faced letters for vectors and matrices. For a vector $\boldsymbol{x}$, denote by $\boldsymbol{x}[i]$ its $i$-th coordinate. For a matrix $\boldsymbol{A}$, we use $\boldsymbol{A}[i,j]$ to denote its $(i,j)$-th entry, and use $\boldsymbol{A}[i,:]$ and $\boldsymbol{A}[:,j]$ to denote its $i$-th row and $j$-th column, respectively (both as column vectors). We use $\|\cdot\|_2$ or $\|\cdot\|$ to denote the Euclidean norm of a vector, and use $\|\cdot\|_F$ to denote the Frobenius norm of a matrix. We use $\langle \cdot, \cdot \rangle$ to denote the standard Euclidean inner product between two vectors or two matrices. Let $[n] = \{1, 2, \ldots, n\}$.

## 2 The Auto-Balancing Properties in Deep Neural Networks

In this section we study the implicit regularization imposed by gradient descent with infinitesimal step size (gradient flow) in training deep neural networks. In Section 2.1 we consider fully connected neural networks, and our main result (Theorem 2.1) shows that gradient flow automatically balances the incoming and outgoing weights at every neuron. This directly implies that the weights between different layers are balanced (Corollary 2.1). For linear activation, we derive a stronger auto-balancing property (Theorem 2.2). In Section 2.2 we generalize our result from fully connected neural networks to convolutional neural networks. In Section 2.3 we present the proof of Theorem 2.1. The proofs of other theorems in this section follow similar ideas and are deferred to Appendix A.

### 2.1 Fully Connected Neural Networks

We first formally define a fully connected feed-forward neural network with $N$ ($N \geqslant 2$) layers. Let $\boldsymbol{W}^{(h)} \in \mathbb{R}^{n_h \times n_{h-1}}$ be the weight matrix in the $h$-th layer, and define $\boldsymbol{w} = (\boldsymbol{W}^{(h)})_{h=1}^{N}$ as a shorthand of the collection of all the weights. Then the function $f_{\boldsymbol{w}} : \mathbb{R}^d \to \mathbb{R}^p$ ($d = n_0, p = n_N$) computed by this network can be defined recursively: $f_{\boldsymbol{w}}^{(1)}(\boldsymbol{x}) = \boldsymbol{W}^{(1)}\boldsymbol{x}$, $f_{\boldsymbol{w}}^{(h)}(\boldsymbol{x}) = \boldsymbol{W}^{(h)}\phi_{h-1}(f_{\boldsymbol{w}}^{(h-1)}(\boldsymbol{x}))$ ($h = 2, \ldots, N$), and $f_{\boldsymbol{w}}(\boldsymbol{x}) = f_{\boldsymbol{w}}^{(N)}(\boldsymbol{x})$, where each $\phi_h$ is an activation function that acts coordinate-wise on vectors.[8] We assume that each $\phi_h$ ($h \in [N-1]$) is *homogeneous*, namely, $\phi_h(x) = \phi_h'(x) \cdot x$ for all $x$ and all elements of the sub-differential $\phi_h'(\cdot)$ when $\phi_h$ is non-differentiable at $x$. This property is satisfied by functions like ReLU $\phi(x) = \max\{x, 0\}$, Leaky ReLU $\phi(x) = \max\{x, \alpha x\}$ ($0 < \alpha < 1$), and linear function $\phi(x) = x$.

Let $\ell : \mathbb{R}^p \times \mathbb{R}^p \to \mathbb{R}_{\geqslant 0}$ be a differentiable loss function. Given a training dataset $\{(\boldsymbol{x}_i, \boldsymbol{y}_i)\}_{i=1}^{m} \subset \mathbb{R}^d \times \mathbb{R}^p$, the training loss as a function of the network parameters $\boldsymbol{w}$ is defined as

$$L(\boldsymbol{w}) = \frac{1}{m} \sum_{i=1}^{m} \ell\left(f_{\boldsymbol{w}}(\boldsymbol{x}_i), \boldsymbol{y}_i\right). \tag{4}$$

We consider gradient descent with infinitesimal step size (also known as gradient flow) applied on $L(\boldsymbol{w})$, which is captured by the differential inclusion:

$$\frac{\mathrm{d}\boldsymbol{W}^{(h)}}{\mathrm{d}t} \in -\frac{\partial L(\boldsymbol{w})}{\partial \boldsymbol{W}^{(h)}}, \qquad h = 1, \ldots, N, \tag{5}$$

where $t$ is a continuous time index, and $\frac{\partial L(\boldsymbol{w})}{\partial \boldsymbol{W}^{(h)}}$ is the Clarke sub-differential [Clarke et al., 2008]. If curves $\boldsymbol{W}^{(h)} = \boldsymbol{W}^{(h)}(t)$ ($h \in [N]$) evolve with time according to (5) they are said to be a solution of the gradient flow differential inclusion.

Our main result in this section is the following invariance imposed by gradient flow.

**Theorem 2.1** (Balanced incoming and outgoing weights at every neuron). *For any $h \in [N-1]$ and $i \in [n_h]$, we have*

$$\frac{\mathrm{d}}{\mathrm{d}t}\left(\|\boldsymbol{W}^{(h)}[i,:]\|^2 - \|\boldsymbol{W}^{(h+1)}[:,i]\|^2\right) = 0. \tag{6}$$

Note that $\boldsymbol{W}^{(h)}[i,:]$ is a vector consisting of network weights coming into the $i$-th neuron in the $h$-th hidden layer, and $\boldsymbol{W}^{(h+1)}[:,i]$ is the vector of weights going out from the same neuron. Therefore, Theorem 2.1 shows that gradient flow exactly preserves the difference between the squared $\ell_2$-norms of incoming weights and outgoing weights at any neuron.

Taking sum of (6) over $i \in [n_h]$, we obtain the following corollary which says gradient flow preserves the difference between the squares of Frobenius norms of weight matrices.

**Corollary 2.1** (Balanced weights across layers). *For any $h \in [N-1]$, we have*

$$\frac{\mathrm{d}}{\mathrm{d}t}\left(\|\boldsymbol{W}^{(h)}\|_F^2 - \|\boldsymbol{W}^{(h+1)}\|_F^2\right) = 0.$$

Corollary 2.1 explains why in practice, trained multi-layer models usually have similar magnitudes on all the layers: if we use a small initialization, $\|\boldsymbol{W}^{(h)}\|_F^2 - \|\boldsymbol{W}^{(h+1)}\|_F^2$ is very small at the beginning, and Corollary 2.1 implies this difference remains small at all time. This finding also partially explains why gradient descent converges. Although the objective function like (4) may not be smooth over the entire parameter space, given that $\|\boldsymbol{W}^{(h)}\|_F^2 - \|\boldsymbol{W}^{(h+1)}\|_F^2$ is small for all $h$, the objective function may have smoothness. Under this condition, standard theory shows that gradient descent converges. We believe this finding serves as a key building block for understanding first order methods for training deep neural networks.

For linear activation, we have the following stronger invariance than Theorem 2.1:

**Theorem 2.2** (Stronger balancedness property for linear activation). *If for some $h \in [N-1]$ we have $\phi_h(x) = x$, then*

$$\frac{\mathrm{d}}{\mathrm{d}t}\left(\boldsymbol{W}^{(h)}(\boldsymbol{W}^{(h)})^\top - (\boldsymbol{W}^{(h+1)})^\top\boldsymbol{W}^{(h+1)}\right) = \mathbf{0}.$$

This result was known for linear networks [Arora et al., 2018], but the proof there relies on the entire network being linear while Theorem 2.2 only needs two consecutive layers to have no nonlinear activations in between.

While Theorem 2.1 shows the invariance in a node-wise manner, Theorem 2.2 shows for linear activation, we can derive a layer-wise invariance. Inspired by this strong invariance, in Section 3 we prove gradient descent with positive step sizes preserves this invariance approximately for matrix factorization.

## 2.2 Convolutional Neural Networks

Now we show that the conservation property in Corollary 2.1 can be generalized to convolutional neural networks. In fact, we can allow *arbitrary sparsity pattern and weight sharing structure* within a layer; convolutional layers are a special case.

**Neural networks with sparse connections and shared weights.** We use the same notation as in Section 2.1, with the difference that some weights in a layer can be *missing* or *shared*. Formally, the weight matrix $\boldsymbol{W}^{(h)} \in \mathbb{R}^{n_h \times n_{h-1}}$ in layer $h$ ($h \in [N]$) can be described by a vector $\boldsymbol{v}^{(h)} \in \mathbb{R}^{d_h}$ and a function $g_h : [n_h] \times [n_{h-1}] \to [d_h] \cup \{0\}$. Here $\boldsymbol{v}^{(h)}$ consists of the actual *free parameters* in this layer and $d_h$ is the number of free parameters (e.g. if there are $k$ convolutional filters in layer $h$ each with size $r$, we have $d_h = r \cdot k$). The map $g_h$ represents the sparsity and weight sharing pattern:

$$\boldsymbol{W}^{(h)}[i,j] = \begin{cases} 0, & g_h(i,j) = 0, \\ \boldsymbol{v}^{(h)}[k], & g_h(i,j) = k > 0. \end{cases}$$

Denote by $\boldsymbol{v} = \left(\boldsymbol{v}^{(h)}\right)_{h=1}^{N}$ the collection of all the parameters in this network, and we consider gradient flow to learn the parameters:

$$\frac{\mathrm{d}\boldsymbol{v}^{(h)}}{\mathrm{d}t} \in -\frac{\partial L(\boldsymbol{v})}{\partial \boldsymbol{v}^{(h)}}, \qquad h = 1, \dots, N.$$

The following theorem generalizes Corollary 2.1 to neural networks with sparse connections and shared weights:

**Theorem 2.3.** *For any $h \in [N-1]$, we have*

$$\frac{\mathrm{d}}{\mathrm{d}t}\left(\|\boldsymbol{v}^{(h)}\|^2 - \|\boldsymbol{v}^{(h+1)}\|^2\right) = 0.$$

Therefore, for a neural network with arbitrary sparsity pattern and weight sharing structure, gradient flow still balances the magnitudes of all layers.

## 2.3 Proof of Theorem 2.1

The proofs of all theorems in this section are similar. They are based on the use of the chain rule (i.e. back-propagation) and the property of homogeneous activations. Below we provide the proof of Theorem 2.1 and defer the proofs of other theorems to Appendix A.

*Proof of Theorem 2.1.* First we note that we can without loss of generality assume $L$ is the loss associated with one data sample $(\boldsymbol{x}, \boldsymbol{y}) \in \mathbb{R}^d \times \mathbb{R}^p$, i.e., $L(\boldsymbol{w}) = \ell(f_{\boldsymbol{w}}(\boldsymbol{x}), \boldsymbol{y})$. In fact, for $L(\boldsymbol{w}) = \frac{1}{m}\sum_{k=1}^m L_k(\boldsymbol{w})$ where $L_k(\boldsymbol{w}) = \ell\left(f_{\boldsymbol{w}}(\boldsymbol{x}_k), \boldsymbol{y}_k\right)$, for any single weight $\boldsymbol{W}^{(h)}[i,j]$ in the network we can compute $\frac{\mathrm{d}}{\mathrm{d}t}(\boldsymbol{W}^{(h)}[i,j])^2 = 2\boldsymbol{W}^{(h)}[i,j] \cdot \frac{\mathrm{d}\boldsymbol{W}^{(h)}[i,j]}{\mathrm{d}t} = -2\boldsymbol{W}^{(h)}[i,j] \cdot \frac{\partial L(\boldsymbol{w})}{\partial \boldsymbol{W}^{(h)}[i,j]} = -2\boldsymbol{W}^{(h)}[i,j] \cdot \frac{1}{m}\sum_{k=1}^m \frac{\partial L_k(\boldsymbol{w})}{\partial \boldsymbol{W}^{(h)}[i,j]}$, using the sharp chain rule of differential inclusions for tame functions [Drusvyatskiy et al., 2015, Davis et al., 2018]. Thus, if we can prove the theorem for every individual loss $L_k$, we can prove the theorem for $L$ by taking average over $k \in [m]$.

Therefore in the rest of proof we assume $L(\boldsymbol{w}) = \ell(f_{\boldsymbol{w}}(\boldsymbol{x}), \boldsymbol{y})$. For convenience, we denote $\boldsymbol{x}^{(h)} = f_{\boldsymbol{w}}^{(h)}(\boldsymbol{x})$ ($h \in [N]$), which is the input to the $h$-th hidden layer of neurons for $h \in [N-1]$ and is the output of the network for $h = N$. We also denote $\boldsymbol{x}^{(0)} = \boldsymbol{x}$ and $\phi_0(x) = x$ ($\forall x$).

Now we prove (6). Since $\boldsymbol{W}^{(h+1)}[k,i]$ ($k \in [n_{h+1}]$) can only affect $L(\boldsymbol{w})$ through $\boldsymbol{x}^{(h+1)}[k]$, we have for $k \in [n_{h+1}]$,

$$\frac{\partial L(\boldsymbol{w})}{\partial \boldsymbol{W}^{(h+1)}[k,i]} = \frac{\partial L(\boldsymbol{w})}{\partial \boldsymbol{x}^{(h+1)}[k]} \cdot \frac{\partial \boldsymbol{x}^{(h+1)}[k]}{\partial \boldsymbol{W}^{(h+1)}[k,i]} = \frac{\partial L(\boldsymbol{w})}{\partial \boldsymbol{x}^{(h+1)}[k]} \cdot \phi_h(\boldsymbol{x}^{(h)}[i]),$$

which can be rewritten as

$$\frac{\partial L(\boldsymbol{w})}{\partial \boldsymbol{W}^{(h+1)}[:,i]} = \phi_h(\boldsymbol{x}^{(h)}[i]) \cdot \frac{\partial L(\boldsymbol{w})}{\partial \boldsymbol{x}^{(h+1)}}.$$

It follows that

$$\frac{\mathrm{d}}{\mathrm{d}t}\|\boldsymbol{W}^{(h+1)}[:,i]\|^2 = 2\left\langle \boldsymbol{W}^{(h+1)}[:,i], \frac{\mathrm{d}}{\mathrm{d}t}\boldsymbol{W}^{(h+1)}[:,i]\right\rangle = -2\left\langle \boldsymbol{W}^{(h+1)}[:,i], \frac{\partial L(\boldsymbol{w})}{\partial \boldsymbol{W}^{(h+1)}[:,i]}\right\rangle$$

$$= -2\phi_h(\boldsymbol{x}^{(h)}[i]) \cdot \left\langle \boldsymbol{W}^{(h+1)}[:,i], \frac{\partial L(\boldsymbol{w})}{\partial \boldsymbol{x}^{(h+1)}}\right\rangle.$$

$$(7)$$

On the other hand, $\boldsymbol{W}^{(h)}[i,:]$ only affects $L(\boldsymbol{w})$ through $\boldsymbol{x}^{(h)}[i]$. Using the chain rule, we get

$$\frac{\partial L(\boldsymbol{w})}{\partial \boldsymbol{W}^{(h)}[i,:]} = \frac{\partial L(\boldsymbol{w})}{\partial \boldsymbol{x}^{(h)}[i]} \cdot \phi_{h-1}(\boldsymbol{x}^{(h-1)}) = \left\langle \frac{\partial L(\boldsymbol{w})}{\partial \boldsymbol{x}^{(h+1)}}, \boldsymbol{W}^{(h+1)}[:,i]\right\rangle \cdot \phi_h'(\boldsymbol{x}^{(h)}[i]) \cdot \phi_{h-1}(\boldsymbol{x}^{(h-1)}),$$

where $\phi'$ is interpreted as a set-valued mapping whenever it is applied at a non-differentiable point.[9]

It follows that[10]

$$\frac{\mathrm{d}}{\mathrm{d}t}\|\boldsymbol{W}^{(h)}[i,:]\|^2 = 2\left\langle \boldsymbol{W}^{(h)}[i,:], \frac{\mathrm{d}}{\mathrm{d}t}\boldsymbol{W}^{(h)}[i,:]\right\rangle = -2\left\langle \boldsymbol{W}^{(h)}[i,:], \frac{\partial L(\boldsymbol{w})}{\partial \boldsymbol{W}^{(h)}[i,:]}\right\rangle$$

$$= -2\left\langle \frac{\partial L(\boldsymbol{w})}{\partial \boldsymbol{x}^{(h+1)}}, \boldsymbol{W}^{(h+1)}[:,i]\right\rangle \cdot \phi_h'(\boldsymbol{x}^{(h)}[i]) \cdot \left\langle \boldsymbol{W}^{(h)}[i,:], \phi_{h-1}(\boldsymbol{x}^{(h-1)})\right\rangle$$

$$= -2\left\langle \frac{\partial L(\boldsymbol{w})}{\partial \boldsymbol{x}^{(h+1)}}, \boldsymbol{W}^{(h+1)}[:,i]\right\rangle \cdot \phi_h'(\boldsymbol{x}^{(h)}[i]) \cdot \boldsymbol{x}^{(h)}[i] = -2\left\langle \frac{\partial L(\boldsymbol{w})}{\partial \boldsymbol{x}^{(h+1)}}, \boldsymbol{W}^{(h+1)}[:,i]\right\rangle \cdot \phi_h(\boldsymbol{x}^{(h)}[i]).$$

Comparing the above expression to (7), we finish the proof. $\square$

## 3 Gradient Descent Converges to Global Minimum for Asymmetric Matrix Factorization

In this section we constrain ourselves to the asymmetric matrix factorization problem and analyze the gradient descent algorithm with random initialization. Our analysis is inspired by the auto-balancing properties presented in Section 2. We extend these properties from gradient flow to gradient descent with positive step size.

Formally, we study the following non-convex optimization problem:

$$\min_{\boldsymbol{U}\in\mathbb{R}^{d_1\times r}, \boldsymbol{V}\in\mathbb{R}^{d_2\times r}} f(\boldsymbol{U},\boldsymbol{V}) = \frac{1}{2}\left\|\boldsymbol{U}\boldsymbol{V}^\top - \boldsymbol{M}^*\right\|_F^2, \tag{8}$$

where $\boldsymbol{M}^* \in \mathbb{R}^{d_1\times d_2}$ has rank $r$. Note that we do not have any explicit regularization in (8). The gradient descent dynamics for (8) have the following form:

$$\boldsymbol{U}_{t+1} = \boldsymbol{U}_t - \eta_t(\boldsymbol{U}_t\boldsymbol{V}_t^\top - \boldsymbol{M}^*)\boldsymbol{V}_t, \qquad \boldsymbol{V}_{t+1} = \boldsymbol{V}_t - \eta_t(\boldsymbol{U}_t\boldsymbol{V}_t^\top - \boldsymbol{M}^*)^\top\boldsymbol{U}_t. \tag{9}$$

### 3.1 The General Rank-$r$ Case

First we consider the general case of $r \geqslant 1$. Our main theorem below says that if we use a random small initialization $(\boldsymbol{U}_0, \boldsymbol{V}_0)$, and set step sizes $\eta_t$ to be appropriately small, then gradient descent (9) will converge to a solution close to the global minimum of (8). To our knowledge, this is the first result showing that gradient descent with random initialization directly solves the un-regularized asymmetric matrix factorization problem (8).

**Theorem 3.1.** *Let $0 < \epsilon < \|\boldsymbol{M}^*\|_F$. Suppose we initialize the entries in $\boldsymbol{U}_0$ and $\boldsymbol{V}_0$ i.i.d. from $\mathcal{N}(0, \frac{\epsilon}{\mathrm{poly}(d)})$ ($d = \max\{d_1, d_2\}$), and run (9) with step sizes $\eta_t = \frac{\sqrt{\epsilon/r}}{100(t+1)\|\boldsymbol{M}^*\|_F^{3/2}}$ ($t = 0, 1, \ldots$).[11] Then with high probability over the initialization, $\lim_{t\to\infty}(\boldsymbol{U}_t, \boldsymbol{V}_t) = (\bar{\boldsymbol{U}}, \bar{\boldsymbol{V}})$ exists and satisfies $\left\|\bar{\boldsymbol{U}}\bar{\boldsymbol{V}}^\top - \boldsymbol{M}^*\right\|_F \leqslant \epsilon$.*

**Proof sketch of Theorem 3.1.** First let's imagine that we are using infinitesimal step size in GD. Then according to Theorem 2.2 (viewing problem (8) as learning a two-layer linear network where the inputs are all the standard unit vectors in $\mathbb{R}^{d_2}$), we know that $\boldsymbol{U}^\top\boldsymbol{U} - \boldsymbol{V}^\top\boldsymbol{V}$ will stay invariant throughout the algorithm. Hence when $\boldsymbol{U}$ and $\boldsymbol{V}$ are initialized to be small, $\boldsymbol{U}^\top\boldsymbol{U} - \boldsymbol{V}^\top\boldsymbol{V}$ will stay small forever. Combined with the fact that the objective $f(\boldsymbol{U}, \boldsymbol{V})$ is decreasing over time (which means $\boldsymbol{U}\boldsymbol{V}^\top$ cannot be too far from $\boldsymbol{M}^*$), we can show that $\boldsymbol{U}$ and $\boldsymbol{V}$ will always stay bounded.

Now we are using positive step sizes $\eta_t$, so we no longer have the invariance of $\boldsymbol{U}^\top\boldsymbol{U} - \boldsymbol{V}^\top\boldsymbol{V}$. Nevertheless, by a careful analysis of the updates, we can still prove that $\boldsymbol{U}_t^\top\boldsymbol{U}_t - \boldsymbol{V}_t^\top\boldsymbol{V}_t$ is small, the objective $f(\boldsymbol{U}_t, \boldsymbol{V}_t)$ decreases, and $\boldsymbol{U}_t$ and $\boldsymbol{V}_t$ stay bounded. Formally, we have the following lemma:

**Lemma 3.1.** *With high probability over the initialization $(\boldsymbol{U}_0, \boldsymbol{V}_0)$, for all $t$ we have:*

(i) *Balancedness:* $\left\| \boldsymbol{U}_t^\top \boldsymbol{U}_t - \boldsymbol{V}_t^\top \boldsymbol{V}_t \right\|_F \leqslant \epsilon;$

(ii) *Decreasing objective:* $f(\boldsymbol{U}_t, \boldsymbol{V}_t) \leqslant f(\boldsymbol{U}_{t-1}, \boldsymbol{V}_{t-1}) \leqslant \cdots \leqslant f(\boldsymbol{U}_0, \boldsymbol{V}_0) \leqslant 2 \left\| \boldsymbol{M}^* \right\|_F^2;$

(iii) *Boundedness:* $\left\| \boldsymbol{U}_t \right\|_F^2 \leqslant 5\sqrt{r} \left\| \boldsymbol{M}^* \right\|_F, \left\| \boldsymbol{V}_t \right\|_F^2 \leqslant 5\sqrt{r} \left\| \boldsymbol{M}^* \right\|_F.$

Now that we know the GD algorithm automatically constrains $(\boldsymbol{U}_t, \boldsymbol{V}_t)$ in a bounded region, we can use the smoothness of $f$ in this region and a standard analysis of GD to show that $(\boldsymbol{U}_t, \boldsymbol{V}_t)$ converges to a stationary point $(\bar{\boldsymbol{U}}, \bar{\boldsymbol{V}})$ of $f$ (Lemma B.2). Furthermore, using the results of [Lee et al., 2016, Panageas and Piliouras, 2016] we know that $(\bar{\boldsymbol{U}}, \bar{\boldsymbol{V}})$ is almost surely not a strict saddle point. Then the following lemma implies that $(\bar{\boldsymbol{U}}, \bar{\boldsymbol{V}})$ has to be close to a global optimum since we know $\left\| \bar{\boldsymbol{U}}^\top \bar{\boldsymbol{U}} - \bar{\boldsymbol{V}}^\top \bar{\boldsymbol{V}} \right\|_F \leqslant \epsilon$ from Lemma 3.1 (i). This would complete the proof of Theorem 3.1.

**Lemma 3.2.** *Suppose* $(\boldsymbol{U}, \boldsymbol{V})$ *is a stationary point of* $f$ *such that* $\left\| \boldsymbol{U}^\top \boldsymbol{U} - \boldsymbol{V}^\top \boldsymbol{V} \right\|_F \leqslant \epsilon.$ *Then either* $\left\| \boldsymbol{U}\boldsymbol{V}^\top - \boldsymbol{M}^* \right\|_F \leqslant \epsilon,$ *or* $(\boldsymbol{U}, \boldsymbol{V})$ *is a strict saddle point of* $f.$

The full proof of Theorem 3.1 and the proofs of Lemmas 3.1 and 3.2 are given in Appendix B.

### 3.2 The Rank-1 Case

We have shown in Theorem 3.1 that GD with small and diminishing step sizes converges to a global minimum for matrix factorization. Empirically, it is observed that a constant step size $\eta_t \equiv \eta$ is enough for GD to converge quickly to global minimum. Therefore, some natural questions are how to prove convergence of GD with a constant step size, how fast it converges, and how the discretization affects the invariance we derived in Section 2.

While these questions remain challenging for the general rank-$r$ matrix factorization, we resolve them for the case of $r = 1$. Our main finding is that with constant step size, the norms of two layers are always within a constant factor of each other (although we may no longer have the stronger balancedness property as in Lemma 3.1), and we utilize this property to prove the *linear convergence* of GD to a global minimum.

When $r = 1$, the asymmetric matrix factorization problem and its GD dynamics become

$$\min_{\boldsymbol{u} \in \mathbb{R}^{d_1}, \boldsymbol{v} \in \mathbb{R}^{d_2}} \frac{1}{2} \left\| \boldsymbol{u}\boldsymbol{v}^\top - \boldsymbol{M}^* \right\|_F^2$$

and

$$\boldsymbol{u}_{t+1} = \boldsymbol{u}_t - \eta(\boldsymbol{u}_t \boldsymbol{v}_t^\top - \boldsymbol{M}^*)\boldsymbol{v}_t, \qquad \boldsymbol{v}_{t+1} = \boldsymbol{v}_t - \eta \left( \boldsymbol{v}_t \boldsymbol{u}_t^\top - \boldsymbol{M}^{*\top} \right) \boldsymbol{u}_t.$$

Here we assume $\boldsymbol{M}^*$ has rank 1, i.e., it can be factorized as $\boldsymbol{M}^* = \sigma_1 \boldsymbol{u}^* \boldsymbol{v}^{*\top}$ where $\boldsymbol{u}^*$ and $\boldsymbol{v}^*$ are unit vectors and $\sigma_1 > 0$.

Our main theoretical result is the following.

**Theorem 3.2** (Approximate balancedness and linear convergence of GD for rank-1 matrix factorization)**.** *Suppose* $\boldsymbol{u}_0 \sim \mathcal{N}(\mathbf{0}, \delta \boldsymbol{I}),$ $\boldsymbol{v}_0 \sim \mathcal{N}(\mathbf{0}, \delta \boldsymbol{I})$ *with* $\delta = c_{init} \sqrt{\frac{\sigma_1}{d}}$ $(d = \max\{d_1, d_2\})$ *for some sufficiently small constant* $c_{init} > 0,$ *and* $\eta = \frac{c_{step}}{\sigma_1}$ *for some sufficiently small constant* $c_{step} > 0.$ *Then with constant probability over the initialization, for all* $t$ *we have* $c_0 \leqslant \frac{|\boldsymbol{u}_t^\top \boldsymbol{u}^*|}{|\boldsymbol{v}_t^\top \boldsymbol{v}^*|} \leqslant C_0$ *for some universal constants* $c_0, C_0 > 0.$ *Furthermore, for any* $0 < \epsilon < 1,$ *after* $t = O \left( \log \frac{d}{\epsilon} \right)$ *iterations, we have* $\left\| \boldsymbol{u}_t \boldsymbol{v}_t^\top - \boldsymbol{M}^* \right\|_F \leqslant \epsilon \sigma_1.$

Theorem 3.2 shows for $\boldsymbol{u}_t$ and $\boldsymbol{v}_t$, their strengths in the signal space, $|\boldsymbol{u}_t^\top \boldsymbol{u}^*|$ and $|\boldsymbol{v}_t^\top \boldsymbol{v}^*|$, are of the same order. This approximate balancedness helps us prove the linear convergence of GD. We refer readers to Appendix C for the proof of Theorem 3.2.

## 4 Empirical Verification

We perform experiments to verify the auto-balancing properties of gradient descent in neural networks with ReLU activation. Our results below show that for GD with small step size and small

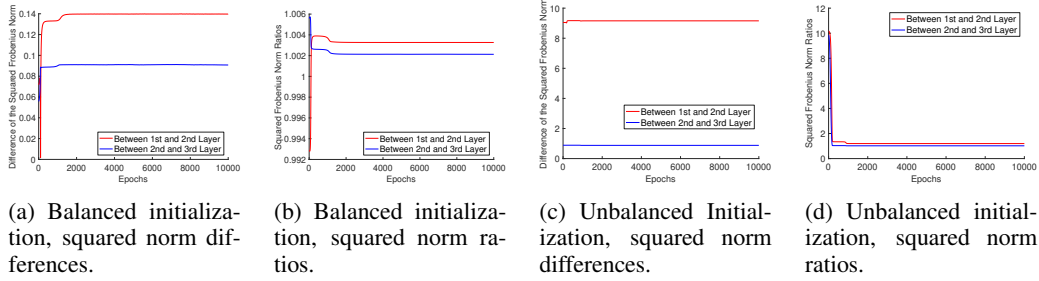

(a) Balanced initialization, squared norm differences.

(b) Balanced initialization, squared norm ratios.

(c) Unbalanced Initialization, squared norm differences.

(d) Unbalanced initialization, squared norm ratios.

Figure 2: Balancedness of a 3-layer neural network.

initialization: (1) the difference between the squared Frobenius norms of any two layers remains small in all iterations, and (2) the ratio between the squared Frobenius norms of any two layers becomes close to 1. Notice that our theorems in Section 2 hold for gradient flow (step size $\to 0$) but in practice we can only choose a (small) positive step size, so we cannot hope the difference between the squared Frobenius norms to remain exactly the same but can only hope to observe that the differences remain small.

We consider a 3-layer fully connected network of the form $f(x) = W_3\phi(W_2\phi(W_1 x))$ where $x \in \mathbb{R}^{1,000}$ is the input, $W_1 \in \mathbb{R}^{100 \times 1,000}$, $W_2 \in \mathbb{R}^{100 \times 100}$, $W_3 \in \mathbb{R}^{10 \times 100}$, and $\phi(\cdot)$ is ReLU activation. We use 1,000 data points and the quadratic loss function, and run GD. We first test a balanced initialization: $W_1[i,j] \sim N(0, \frac{10^{-4}}{100})$, $W_2[i,j] \sim N(0, \frac{10^{-4}}{10})$ and $W_3[i,j] \sim N(0, 10^{-4})$, which ensures $\|W_1\|_F^2 \approx \|W_2\|_F^2 \approx \|W_3\|_F^2$. After 10,000 iterations we have $\|W_1\|_F^2 = 42.90$, $\|W_2\|_F^2 = 43.76$ and $\|W_3\|_F^2 = 43.68$. Figure 2a shows that in all iterations $\left|\|W_1\|_F^2 - \|W_2\|_F^2\right|$ and $\left|\|W_2\|_F^2 - \|W_3\|_F^2\right|$ are bounded by 0.14 which is much smaller than the magnitude of each $\|W_h\|_F^2$. Figures 2b shows that the ratios between norms approach 1. We then test an unbalanced initialization: $W_1[i,j] \sim N(0, 10^{-4})$, $W_2[i,j] \sim N(0, 10^{-4})$ and $W_3[i,j] \sim N(0, 10^{-4})$. After 10,000 iterations we have $\|W_1\|_F^2 = 55.50$, $\|W_2\|_F^2 = 45.65$ and $\|W_3\|_F^2 = 45.46$. Figure 2c shows that $\left|\|W_1\|_F^2 - \|W_2\|_F^2\right|$ and $\left|\|W_2\|_F^2 - \|W_3\|_F^2\right|$ are bounded by 9 (and indeed change very little throughout the process), and Figures 2d shows that the ratios become close to 1 after about 1,000 iterations.

## 5    Conclusion and Future Work

In this paper we take a step towards characterizing the invariance imposed by first order algorithms. We show that gradient flow automatically balances the magnitudes of all layers in a deep neural network with homogeneous activations. For the concrete model of asymmetric matrix factorization, we further use the balancedness property to show that gradient descent converges to global minimum. We believe our findings on the invariance in deep models could serve as a fundamental building block for understanding optimization in deep learning. Below we list some future directions.

**Other first-order methods.**   In this paper we focus on the invariance induced by gradient descent. In practice, different acceleration and adaptive methods are also used. A natural future direction is how to characterize the invariance properties of these algorithms.

**From gradient flow to gradient descent: a generic analysis?**   As discussed in Section 3, while strong invariance properties hold for gradient flow, in practice one uses gradient descent with positive step sizes and the invariance may only hold approximately because positive step sizes discretize the dynamics. We use specialized techniques for analyzing asymmetric matrix factorization. It would be very interesting to develop a generic approach to analyze the discretization. Recent findings on the connection between optimization and ordinary differential equations [Su et al., 2014, Zhang et al., 2018] might be useful for this purpose.

## Acknowledgements

We thank Phil Long for his helpful comments on an earlier draft of this paper. JDL acknowledges support from ARO W911NF-11-1-0303.

## Footnotes

*The full version of this paper is available at https://arxiv.org/abs/1806.00900.

[5]A function $f$ is coercive if $\|\boldsymbol{x}\| \to \infty$ implies $f(\boldsymbol{x}) \to \infty$.

[6]A function is said to be smooth if its gradient is $\beta$-Lipschitz continuous for some finite $\beta > 0$.

[7]A saddle point of a function $f$ is strict if the Hessian at that point has a negative eigenvalue.

[8]We omit the trainable bias weights in the network for simplicity, but our results can be directly generalized to allow bias weights.

[9]More precisely, the equalities should be an inclusion whenever there is a sub-differential, but as we see in the next display the ambiguity in the choice of sub-differential does not affect later calculations.

[10]This holds for any choice of element of the sub-differential, since $\phi'(x)x = \phi(x)$ holds at $x = 0$ for any choice of sub-differential.

[11]The dependency of $\eta_t$ on $t$ can be $\eta_t = \Theta\left(t^{-(1/2+\delta)}\right)$ for any constant $\delta \in (0, 1/2]$.

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
