[Supplementary Material]

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

{\left| \boldsymbol{u}_t^\top \boldsymbol{u}^* \right|}{\left| \boldsymbol{v}_t^\top \boldsymbol{v}^* \right|} \leqslant C_0$ *for some universal constants* $c_0, C_0 > 0$. *Furthermore, for any $0 < \epsilon < 1$, after $t = O\left(\log \frac{d}{\epsilon}\right)$ iterations, we have* $\left\| \boldsymbol{u}_t \boldsymbol{v}_t^\top - \boldsymbol{M}^* \right\|_F \leqslant \epsilon \sigma_1$.

Theorem 3.2 shows for $\boldsymbol{u}_t$ and $\boldsymbol{v}_t$, their strengths in the signal space, $\left| \boldsymbol{u}_t^\top \boldsymbol{u}^* \right|$ and $\left| \boldsymbol{v}_t^\top \boldsymbol{v}^* \right|$,

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

# Appendix

## A    Proofs for Section 2

*Proof of Theorem 2.2.* Same as the proof of Theorem 2.1, we assume without loss of generality that $L(\boldsymbol{w}) = \ell(f_{\boldsymbol{w}}(\boldsymbol{x}), \boldsymbol{y})$ for some $(\boldsymbol{x}, \boldsymbol{y}) \in \mathbb{R}^d \times \mathbb{R}^p$. We also denote $\boldsymbol{x}^{(h)} = f_{\boldsymbol{w}}^{(h)}(\boldsymbol{x})$ ($\forall h \in [N]$), $\boldsymbol{x}^{(0)} = \boldsymbol{x}$ and $\phi_0(x) = x$.

Now we suppose $\phi_h(x) = x$ for some $h \in [N - 1]$. Denote $\boldsymbol{u} = \phi_{h-1}(\boldsymbol{x}^{(h-1)})$. Then we have $\boldsymbol{x}^{(h+1)} = \boldsymbol{W}^{(h+1)}\boldsymbol{x}^{(h)} = \boldsymbol{W}^{(h+1)}\boldsymbol{W}^{(h)}\boldsymbol{u}$. Using the chain rule, we can directly compute

$$\frac{\partial L(\boldsymbol{w})}{\partial \boldsymbol{W}^{(h)}} = \frac{\partial L(\boldsymbol{w})}{\partial \boldsymbol{x}^{(h)}}\boldsymbol{u}^\top = (\boldsymbol{W}^{(h+1)})^\top \frac{\partial L(\boldsymbol{w})}{\partial \boldsymbol{x}^{(h+1)}}\boldsymbol{u}^\top,$$

$$\frac{\partial L(\boldsymbol{w})}{\partial \boldsymbol{W}^{(h+1)}} = \frac{\partial L(\boldsymbol{w})}{\partial \boldsymbol{x}^{(h+1)}}(\boldsymbol{x}^{(h)})^\top = \frac{\partial L(\boldsymbol{w})}{\partial \boldsymbol{x}^{(h+1)}}(\boldsymbol{W}^{(h)}\boldsymbol{u})^\top.$$

Then we have

$$\frac{\mathrm{d}}{\mathrm{d}t}\left(\boldsymbol{W}^{(h)}(\boldsymbol{W}^{(h)})^\top\right) = \boldsymbol{W}^{(h)}\left(\frac{\mathrm{d}}{\mathrm{d}t}\boldsymbol{W}^{(h)}\right)^\top + \left(\frac{\mathrm{d}}{\mathrm{d}t}\boldsymbol{W}^{(h)}\right)(\boldsymbol{W}^{(h)})^\top$$

$$= \boldsymbol{W}^{(h)}\boldsymbol{u}\left(\frac{\partial L(\boldsymbol{w})}{\partial \boldsymbol{x}^{(h+1)}}\right)^\top \boldsymbol{W}^{(h+1)} + (\boldsymbol{W}^{(h+1)})^\top \frac{\partial L(\boldsymbol{w})}{\partial \boldsymbol{x}^{(h+1)}}\boldsymbol{u}^\top (\boldsymbol{W}^{(h)})^\top,$$

$$\frac{\mathrm{d}}{\mathrm{d}t}\left((\boldsymbol{W}^{(h+1)})^\top \boldsymbol{W}^{(h+1)}\right) = (\boldsymbol{W}^{(h+1)})^\top\left(\frac{\mathrm{d}}{\mathrm{d}t}\boldsymbol{W}^{(h+1)}\right) + \left(\frac{\mathrm{d}}{\mathrm{d}t}\boldsymbol{W}^{(h+1)}\right)^\top \boldsymbol{W}^{(h+1)}$$

$$= (\boldsymbol{W}^{(h+1)})^\top \frac{\partial L(\boldsymbol{w})}{\partial \boldsymbol{x}^{(h+1)}}\boldsymbol{u}^\top (\boldsymbol{W}^{(h)})^\top + \boldsymbol{W}^{(h)}\boldsymbol{u}\left(\frac{\partial L(\boldsymbol{w})}{\partial \boldsymbol{x}^{(h+1)}}\right)^\top \boldsymbol{W}^{(h+1)}.$$

Comparing the above two equations we know $\frac{\mathrm{d}}{\mathrm{d}t}\left(\boldsymbol{W}^{(h)}(\boldsymbol{W}^{(h)})^\top - (\boldsymbol{W}^{(h+1)})^\top \boldsymbol{W}^{(h+1)}\right) = \boldsymbol{0}$.    $\square$

*Proof of Theorem 2.3.* Same as the proof of Theorem 2.1, we assume without loss of generality that $L(\boldsymbol{v}) = L(\boldsymbol{w}) = \ell(f_{\boldsymbol{w}}(\boldsymbol{x}), \boldsymbol{y})$ for $(\boldsymbol{x}, \boldsymbol{y}) \in \mathbb{R}^d \times \mathbb{R}^p$, and denote $\boldsymbol{x}^{(h)} = f_{\boldsymbol{w}}^{(h)}(\boldsymbol{x})$ ($\forall h \in [N]$), $\boldsymbol{x}^{(0)} = \boldsymbol{x}$ and $\phi_0(x) = x$.

Using the chain rule, we have

$$\frac{\partial L(\boldsymbol{v})}{\partial \boldsymbol{v}^{(h+1)}[l]} = \sum_{(k,i):g_{h+1}(k,i)=l} \frac{\partial L(\boldsymbol{v})}{\partial \boldsymbol{x}^{(h+1)}[k]} \cdot \phi_h(\boldsymbol{x}^{(h)}[i]), \qquad l \in [d_{h+1}].$$

Then we have using the sharp chain rule,

$$\frac{\mathrm{d}}{\mathrm{d}t}\|\boldsymbol{v}^{(h+1)}\|^2 = 2\left\langle \boldsymbol{v}^{(h+1)}, \frac{\mathrm{d}}{\mathrm{d}t}\boldsymbol{v}^{(h+1)}\right\rangle = -2\left\langle \boldsymbol{v}^{(h+1)}, \frac{\partial L(\boldsymbol{v})}{\partial \boldsymbol{v}^{(h+1)}}\right\rangle$$

$$= -2\sum_l \sum_{(k,i):g_{h+1}(k,i)=l} \frac{\partial L(\boldsymbol{v})}{\partial \boldsymbol{x}^{(h+1)}[k]} \cdot \boldsymbol{v}^{(h+1)}[l] \cdot \phi_h(\boldsymbol{x}^{(h)}[i])$$

$$= -2\sum_{(k,i)} \frac{\partial L(\boldsymbol{v})}{\partial \boldsymbol{x}^{(h+1)}[k]} \cdot \boldsymbol{W}^{(h+1)}[k, i] \cdot \phi_h(\boldsymbol{x}^{(h)}[i]) \qquad (10)$$

$$= -2\sum_k \frac{\partial L(\boldsymbol{v})}{\partial \boldsymbol{x}^{(h+1)}[k]} \cdot \boldsymbol{x}^{(h+1)}[k]$$

$$= -2\left\langle \frac{\partial L(\boldsymbol{v})}{\partial \boldsymbol{x}^{(h+1)}}, \boldsymbol{x}^{(h+1)}\right\rangle.$$

Substituting $h$ with $h-1$ in (10) gives $\frac{\mathrm{d}}{\mathrm{d}t}\|\boldsymbol{v}^{(h)}\|^2 = -2\left\langle \frac{\partial L(\boldsymbol{v})}{\partial \boldsymbol{x}^{(h)}}, \boldsymbol{x}^{(h)} \right\rangle$, which further implies

$$
\begin{aligned}
\frac{\mathrm{d}}{\mathrm{d}t}\|\boldsymbol{v}^{(h)}\|^2 &= -2\left\langle \frac{\partial L(\boldsymbol{v})}{\partial \boldsymbol{x}^{(h)}}, \boldsymbol{x}^{(h)} \right\rangle = -2\sum_i \frac{\partial L(\boldsymbol{v})}{\partial \boldsymbol{x}^{(h)}[i]} \cdot \boldsymbol{x}^{(h)}[i] \\
&= -2\sum_i \sum_k \frac{\partial L(\boldsymbol{v})}{\partial \boldsymbol{x}^{(h+1)}[k]} \cdot \boldsymbol{W}^{(h+1)}[k,i] \cdot \phi_h'(\boldsymbol{x}^{(h)}[i]) \cdot \boldsymbol{x}^{(h)}[i] \\
&= -2\sum_k \frac{\partial L(\boldsymbol{v})}{\partial \boldsymbol{x}^{(h+1)}[k]} \sum_i \boldsymbol{W}^{(h+1)}[k,i] \cdot \phi_h(\boldsymbol{x}^{(h)}[i]) \qquad (11) \\
&= -2\sum_k \frac{\partial L(\boldsymbol{v})}{\partial \boldsymbol{x}^{(h+1)}[k]} \cdot \boldsymbol{x}^{(h+1)}[k] \\
&= -2\left\langle \frac{\partial L(\boldsymbol{v})}{\partial \boldsymbol{x}^{(h+1)}}, \boldsymbol{x}^{(h+1)} \right\rangle.
\end{aligned}
$$

The proof is finished by combining (10) and (11). $\qquad\square$

## B  Proof for Rank-$r$ Matrix Factorization (Theorem 3.1)

In this section we give the full proof of Theorem 3.1.

First we recall the gradient of our objective function $f(\boldsymbol{U},\boldsymbol{V}) = \frac{1}{2}\|\boldsymbol{U}\boldsymbol{V}^\top - \boldsymbol{M}^*\|_F^2$:

$$
\frac{\partial f(\boldsymbol{U},\boldsymbol{V})}{\partial \boldsymbol{U}} = (\boldsymbol{U}\boldsymbol{V}^\top - \boldsymbol{M}^*)\boldsymbol{V}, \qquad \frac{\partial f(\boldsymbol{U},\boldsymbol{V})}{\partial \boldsymbol{V}} = (\boldsymbol{U}\boldsymbol{V}^\top - \boldsymbol{M}^*)^\top\boldsymbol{U}.
$$

We also need to calculate the Hessian $\nabla^2 f(\boldsymbol{U},\boldsymbol{V})$. The Hessian can be viewed as a matrix that operates on vectorized matrices of dimension $(d_1+d_2)\times r$ (i.e., the same shape as $\begin{pmatrix}\boldsymbol{U}\\\boldsymbol{V}\end{pmatrix}$). Then, for any $\boldsymbol{W}\in\mathbb{R}^{(d_1+d_2)\times r}$, the Hessian $\nabla^2 f(\boldsymbol{W})$ defines a quadratic form

$$
[\nabla^2 f(\boldsymbol{W})](\boldsymbol{A},\boldsymbol{B}) = \sum_{i,j,k,l} \frac{\partial^2 f(\boldsymbol{W})}{\partial \boldsymbol{W}[i,j]\partial \boldsymbol{W}[k,l]}\boldsymbol{A}[i,j]\boldsymbol{B}[k,l], \qquad \forall \boldsymbol{A},\boldsymbol{B}\in\mathbb{R}^{(d_1+d_2)\times r}.
$$

With this notation, we can express the Hessian $\nabla^2 f(\boldsymbol{U},\boldsymbol{V})$ as follows:

$$
\begin{aligned}
[\nabla^2 f(\boldsymbol{U},\boldsymbol{V})](\boldsymbol{\Delta},\boldsymbol{\Delta}) = 2\langle \boldsymbol{U}\boldsymbol{V}^\top - \boldsymbol{M}^*, \boldsymbol{\Delta}_U\boldsymbol{\Delta}_V^\top\rangle + \|\boldsymbol{U}\boldsymbol{\Delta}_V^\top + \boldsymbol{\Delta}_U\boldsymbol{V}^\top\|_F^2, \\
\forall\boldsymbol{\Delta} = \begin{pmatrix}\boldsymbol{\Delta}_U\\\boldsymbol{\Delta}_V\end{pmatrix}, \boldsymbol{\Delta}_U\in\mathbb{R}^{d_1\times r}, \boldsymbol{\Delta}_V\in\mathbb{R}^{d_2\times r}.
\end{aligned}
$$
$$(12)$$

Now we use the expression of the Hessian to prove that $f(\boldsymbol{U},\boldsymbol{V})$ is locally smooth when both arguments $\boldsymbol{U}$ and $\boldsymbol{V}$ are bounded.

**Lemma B.1** (Smoothness over a bounded set). *For any $c > 0$, constrained on the set $\mathcal{S} = \{(\boldsymbol{U},\boldsymbol{V}) : \boldsymbol{U}\in\mathbb{R}^{d_1\times r}, \boldsymbol{V}\in\mathbb{R}^{d_2\times r}, \|\boldsymbol{U}\|_F^2 \leqslant c\|\boldsymbol{M}^*\|_F, \|\boldsymbol{V}\|_F^2 \leqslant c\|\boldsymbol{M}^*\|_F\}$, the function $f$ is $((6c+2)\|\boldsymbol{M}^*\|_F)$-smooth.*

*Proof.* We prove smoothness by giving an upper bound on $\lambda_{\max}(\nabla^2 f(\boldsymbol{U},\boldsymbol{V}))$ for any $(\boldsymbol{U},\boldsymbol{V})\in\mathcal{S}$.

For any $(\boldsymbol{U},\boldsymbol{V})\in\mathcal{S}$ and any $\boldsymbol{\Delta} = \begin{pmatrix}\boldsymbol{\Delta}_U\\\boldsymbol{\Delta}_V\end{pmatrix}$ $(\boldsymbol{\Delta}_U\in\mathbb{R}^{d_1\times r}, \boldsymbol{\Delta}_V\in\mathbb{R}^{d_2\times r})$, from (12) we have

$$
\begin{aligned}
&[\nabla^2 f(\boldsymbol{U},\boldsymbol{V})](\boldsymbol{\Delta},\boldsymbol{\Delta}) \\
&\leqslant 2\|\boldsymbol{U}\boldsymbol{V}^\top - \boldsymbol{M}^*\|_F\|\boldsymbol{\Delta}_U\boldsymbol{\Delta}_V^\top\|_F + \|\boldsymbol{U}\boldsymbol{\Delta}_V^\top + \boldsymbol{\Delta}_U\boldsymbol{V}^\top\|_F^2 \\
&\leqslant 2\left(\|\boldsymbol{U}\|_F\|\boldsymbol{V}^\top\|_F + \|\boldsymbol{M}^*\|_F\right)\|\boldsymbol{\Delta}_U\|_F\|\boldsymbol{\Delta}_V^\top\|_F + \left(\|\boldsymbol{U}\|_F\|\boldsymbol{\Delta}_V^\top\|_F + \|\boldsymbol{\Delta}_U\|_F\|\boldsymbol{V}^\top\|_F\right)^2
\end{aligned}
$$

$$\leqslant 2\left(c\left\|\boldsymbol{M}^*\right\|_F + \left\|\boldsymbol{M}^*\right\|_F\right)\left\|\boldsymbol{\Delta}\right\|_F^2 + \left(2\sqrt{c\left\|\boldsymbol{M}^*\right\|_F}\cdot\left\|\boldsymbol{\Delta}\right\|_F\right)^2$$
$$= (6c+2)\left\|\boldsymbol{M}^*\right\|_F\left\|\boldsymbol{\Delta}\right\|_F^2.$$

This implies $\lambda_{\max}(\nabla^2 f(\boldsymbol{U},\boldsymbol{V})) \leqslant (6c+2)\left\|\boldsymbol{M}^*\right\|_F$. $\qquad\square$

## B.1 Proof of Lemma 3.1

Recall the following three properties we want to prove in Lemma 3.1, which we call $\mathcal{A}(t)$, $\mathcal{B}(t)$ and $\mathcal{C}(t)$, respectively:

$$\begin{aligned}
\mathcal{A}(t): &\quad \left\|\boldsymbol{U}_t^\top\boldsymbol{U}_t - \boldsymbol{V}_t^\top\boldsymbol{V}_t\right\|_F \leqslant \epsilon, \\
\mathcal{B}(t): &\quad f(\boldsymbol{U}_t,\boldsymbol{V}_t) \leqslant f(\boldsymbol{U}_{t-1},\boldsymbol{V}_{t-1}) \leqslant \cdots \leqslant f(\boldsymbol{U}_0,\boldsymbol{V}_0) \leqslant 2\left\|\boldsymbol{M}^*\right\|_F^2, \\
\mathcal{C}(t): &\quad \left\|\boldsymbol{U}_t\right\|_F^2 \leqslant 5\sqrt{r}\left\|\boldsymbol{M}^*\right\|_F, \left\|\boldsymbol{V}_t\right\|_F^2 \leqslant 5\sqrt{r}\left\|\boldsymbol{M}^*\right\|_F.
\end{aligned}$$

We use induction to prove these statements. For $t = 0$, we can make the Gaussian variance in the initialization sufficiently small such that with high probability we have

$$\left\|\boldsymbol{U}_0\right\|_F^2 \leqslant \epsilon, \qquad \left\|\boldsymbol{V}_0\right\|_F^2 \leqslant \epsilon, \qquad \left\|\boldsymbol{U}_0^\top\boldsymbol{U}_0 - \boldsymbol{V}_0^\top\boldsymbol{V}_0\right\|_F \leqslant \frac{\epsilon}{2}.$$

From now on we assume they are all satisfied. Then $\mathcal{A}(0)$ is already satisfied, $\mathcal{C}(0)$ is satisfied because $\epsilon < \left\|\boldsymbol{M}^*\right\|_F$, and $\mathcal{B}(0)$ can be verified by $f(\boldsymbol{U}_0,\boldsymbol{V}_0) = \frac{1}{2}\left\|\boldsymbol{U}_0\boldsymbol{V}_0^\top - \boldsymbol{M}^*\right\|_F^2 \leqslant \left\|\boldsymbol{U}_0\boldsymbol{V}_0^\top\right\|_F^2 + \left\|\boldsymbol{M}^*\right\|_F^2 \leqslant \left\|\boldsymbol{U}_0\right\|_F^2\left\|\boldsymbol{V}_0^\top\right\|_F^2 + \left\|\boldsymbol{M}^*\right\|_F^2 \leqslant \epsilon^2 + \left\|\boldsymbol{M}^*\right\|_F^2 \leqslant 2\left\|\boldsymbol{M}^*\right\|_F^2$.

To prove $\mathcal{A}(t)$, $\mathcal{B}(t)$ and $\mathcal{C}(t)$ for all $t$, we prove the following three claims. Since we have $\mathcal{A}(0)$, $\mathcal{B}(0)$ and $\mathcal{C}(0)$, if the following claims are all true, the proof will be completed by induction.

(i) $\mathcal{B}(0),\ldots,\mathcal{B}(t),\mathcal{C}(0),\ldots,\mathcal{C}(t) \implies \mathcal{A}(t+1)$;

(ii) $\mathcal{B}(0),\ldots,\mathcal{B}(t),\mathcal{C}(t) \implies \mathcal{B}(t+1)$;

(iii) $\mathcal{A}(t),\mathcal{B}(t) \implies \mathcal{C}(t)$.

**Claim B.1.** $\mathcal{B}(0),\ldots,\mathcal{B}(t),\mathcal{C}(0),\ldots,\mathcal{C}(t) \implies \mathcal{A}(t+1)$.

*Proof.* Using the update rule (9) we can calculate

$$\begin{aligned}
&\boldsymbol{U}_{t+1}^\top\boldsymbol{U}_{t+1} - \boldsymbol{V}_{t+1}^\top\boldsymbol{V}_{t+1} \\
&= \left(\boldsymbol{U}_t - \eta_t(\boldsymbol{U}_t\boldsymbol{V}_t^\top - \boldsymbol{M}^*)\boldsymbol{V}_t\right)^\top\left(\boldsymbol{U}_t - \eta_t(\boldsymbol{U}_t\boldsymbol{V}_t^\top - \boldsymbol{M}^*)\boldsymbol{V}_t\right) \\
&\quad - \left(\boldsymbol{V}_t - \eta_t(\boldsymbol{U}_t\boldsymbol{V}_t^\top - \boldsymbol{M}^*)^\top\boldsymbol{U}_t\right)^\top\left(\boldsymbol{V}_t - \eta_t(\boldsymbol{U}_t\boldsymbol{V}_t^\top - \boldsymbol{M}^*)^\top\boldsymbol{U}_t\right) \\
&= \boldsymbol{U}_t^\top\boldsymbol{U}_t - \boldsymbol{V}_t^\top\boldsymbol{V}_t + \eta_t^2\left(\boldsymbol{V}_t^\top\boldsymbol{R}_t^\top\boldsymbol{R}_t\boldsymbol{V}_t - \boldsymbol{U}_t^\top\boldsymbol{R}_t^\top\boldsymbol{R}_t\boldsymbol{U}_t\right),
\end{aligned}$$

where $\boldsymbol{R}_t = \boldsymbol{U}_t\boldsymbol{V}_t^\top - \boldsymbol{M}^*$. Then we have

$$\begin{aligned}
&\left\|\boldsymbol{U}_{t+1}^\top\boldsymbol{U}_{t+1} - \boldsymbol{V}_{t+1}^\top\boldsymbol{V}_{t+1}\right\|_F \\
&\leqslant \left\|\boldsymbol{U}_t^\top\boldsymbol{U}_t - \boldsymbol{V}_t^\top\boldsymbol{V}_t\right\|_F + \eta_t^2\left(\left\|\boldsymbol{V}_t^\top\boldsymbol{R}_t^\top\boldsymbol{R}_t\boldsymbol{V}_t\right\|_F + \left\|\boldsymbol{U}_t^\top\boldsymbol{R}_t^\top\boldsymbol{R}_t\boldsymbol{U}_t\right\|_F\right) \\
&\leqslant \left\|\boldsymbol{U}_t^\top\boldsymbol{U}_t - \boldsymbol{V}_t^\top\boldsymbol{V}_t\right\|_F + \eta_t^2\left(\left\|\boldsymbol{V}_t\right\|_F^2\left\|\boldsymbol{R}_t\right\|_F^2 + \left\|\boldsymbol{U}_t\right\|_F^2\left\|\boldsymbol{R}_t\right\|_F^2\right) \\
&= \left\|\boldsymbol{U}_t^\top\boldsymbol{U}_t - \boldsymbol{V}_t^\top\boldsymbol{V}_t\right\|_F + 2\eta_t^2\left(\left\|\boldsymbol{V}_t\right\|_F^2 + \left\|\boldsymbol{U}_t\right\|_F^2\right)f(\boldsymbol{U}_t,\boldsymbol{V}_t) \\
&\leqslant \left\|\boldsymbol{U}_t^\top\boldsymbol{U}_t - \boldsymbol{V}_t^\top\boldsymbol{V}_t\right\|_F + 2\eta_t^2\cdot 10\sqrt{r}\left\|\boldsymbol{M}^*\right\|_F\cdot 2\left\|\boldsymbol{M}^*\right\|_F^2,
\end{aligned} \tag{13}$$

where the last line is due to $\mathcal{B}(t)$ and $\mathcal{C}(t)$.

Since we have $\mathcal{B}(t')$ and $\mathcal{C}(t')$ for all $t' \leqslant t$, (13) is still true when substituting $t$ with any $t' \leqslant t$. Summing all of them and noting $\left\|\boldsymbol{U}_0^\top\boldsymbol{U}_0 - \boldsymbol{V}_0^\top\boldsymbol{V}_0\right\|_F \leqslant \frac{\epsilon}{2}$, we get

$$\left\|\boldsymbol{U}_{t+1}^\top\boldsymbol{U}_{t+1} - \boldsymbol{V}_{t+1}^\top\boldsymbol{V}_{t+1}\right\|_F$$

$$\leqslant \left\|\boldsymbol{U}_0^\top \boldsymbol{U}_0 - \boldsymbol{V}_0^\top \boldsymbol{V}_0\right\|_F + 40\sqrt{r}\left\|\boldsymbol{M}^*\right\|_F^3 \sum_{i=0}^{t} \eta_i^2$$

$$\leqslant \frac{\epsilon}{2} + 40\sqrt{r}\left\|\boldsymbol{M}^*\right\|_F^3 \sum_{i=0}^{t} \frac{1}{(i+1)^2} \cdot \frac{\epsilon/r}{100^2 \left\|\boldsymbol{M}^*\right\|_F^3}$$

$$\leqslant \epsilon.$$

Therefore we have proved $\mathcal{A}(t+1)$. $\qquad \square$

**Claim B.2.** $\mathcal{B}(0), \ldots, \mathcal{B}(t), \mathcal{C}(t) \implies \mathcal{B}(t+1)$.

*Proof.* Note that we only need to show $f(\boldsymbol{U}_{t+1}, \boldsymbol{V}_{t+1}) \leqslant f(\boldsymbol{U}_t, \boldsymbol{V}_t)$. We prove this using the standard analysis of gradient descent, for which we need the smoothness of the objective function $f$ (Lemma B.1). We first need to bound $\|\boldsymbol{U}_t\|_F$, $\|\boldsymbol{V}_t\|_F$, $\|\boldsymbol{U}_{t+1}\|_F$ and $\|\boldsymbol{V}_{t+1}\|_F$. We know from $\mathcal{C}(t)$ that $\|\boldsymbol{U}_t\|_F^2 \leqslant 5\sqrt{r}\|\boldsymbol{M}^*\|_F$ and $\|\boldsymbol{V}_t\|_F^2 \leqslant 5\sqrt{r}\|\boldsymbol{M}^*\|_F$. We can also bound $\|\boldsymbol{U}_{t+1}\|_F^2$ and $\|\boldsymbol{V}_{t+1}\|_F^2$ easily from the GD update rule:

$$\begin{aligned}
&\|\boldsymbol{U}_{t+1}\|_F^2 \\
&= \left\|\boldsymbol{U}_t - \eta_t(\boldsymbol{U}_t\boldsymbol{V}_t^\top - \boldsymbol{M}^*)\boldsymbol{V}_t\right\|_F^2 \\
&\leqslant 2\|\boldsymbol{U}_t\|_F^2 + 2\eta_t^2 \left\|\boldsymbol{U}_t\boldsymbol{V}_t^\top - \boldsymbol{M}^*\right\|_F^2 \|\boldsymbol{V}_t\|_F^2 \\
&\leqslant 2 \cdot 5\sqrt{r}\|\boldsymbol{M}^*\|_F + 2\eta_t^2 \cdot 2f(\boldsymbol{U}_t, \boldsymbol{V}_t) \cdot 5\sqrt{r}\|\boldsymbol{M}^*\|_F \\
&\leqslant 10\sqrt{r}\|\boldsymbol{M}^*\|_F + 2 \cdot \frac{\epsilon/r}{100^2(t+1)^2\|\boldsymbol{M}^*\|_F^3} \cdot 4\|\boldsymbol{M}^*\|_F^2 \cdot 5\sqrt{r}\|\boldsymbol{M}^*\|_F && \text{(using } \mathcal{B}(t)\text{)} \\
&\leqslant 10\sqrt{r}\|\boldsymbol{M}^*\|_F + \frac{\epsilon}{100} \\
&\leqslant 11\sqrt{r}\|\boldsymbol{M}^*\|_F. && \text{(using } \epsilon < \|\boldsymbol{M}^*\|_F\text{)}
\end{aligned}$$

Let $\beta = (66\sqrt{r} + 2)\|\boldsymbol{M}^*\|_F$. From Lemma B.1, $f$ is $\beta$-smooth over $\mathcal{S} = \{(\boldsymbol{U}, \boldsymbol{V}) : \|\boldsymbol{U}\|_F^2 \leqslant 11\sqrt{r}\|\boldsymbol{M}^*\|_F, \|\boldsymbol{V}\|_F^2 \leqslant 11\sqrt{r}\|\boldsymbol{M}^*\|_F\}$. Also note that $\eta_t < \frac{1}{\beta}$ by our choice. Then using smoothness we have

$$\begin{aligned}
&f(\boldsymbol{U}_{t+1}, \boldsymbol{V}_{t+1}) \\
&\leqslant f(\boldsymbol{U}_t, \boldsymbol{V}_t) + \left\langle \nabla f(\boldsymbol{U}_t, \boldsymbol{V}_t), \begin{pmatrix} \boldsymbol{U}_{t+1} \\ \boldsymbol{V}_{t+1} \end{pmatrix} - \begin{pmatrix} \boldsymbol{U}_t \\ \boldsymbol{V}_t \end{pmatrix} \right\rangle + \frac{\beta}{2}\left\| \begin{pmatrix} \boldsymbol{U}_{t+1} \\ \boldsymbol{V}_{t+1} \end{pmatrix} - \begin{pmatrix} \boldsymbol{U}_t \\ \boldsymbol{V}_t \end{pmatrix} \right\|_F^2 \\
&= f(\boldsymbol{U}_t, \boldsymbol{V}_t) - \eta_t \|\nabla f(\boldsymbol{U}_t, \boldsymbol{V}_t)\|_F^2 + \frac{\beta}{2}\eta_t^2\|\nabla f(\boldsymbol{U}_t, \boldsymbol{V}_t)\|_F^2 \\
&\leqslant f(\boldsymbol{U}_t, \boldsymbol{V}_t) - \frac{\eta_t}{2}\|\nabla f(\boldsymbol{U}_t, \boldsymbol{V}_t)\|_F^2.
\end{aligned} \tag{14}$$

Therefore we have shown $\mathcal{B}(t+1)$. $\qquad \square$

**Claim B.3.** $\mathcal{A}(t), \mathcal{B}(t) \implies \mathcal{C}(t)$.

*Proof.* From $\mathcal{B}(t)$ we know $\frac{1}{2}\left\|\boldsymbol{U}_t\boldsymbol{V}_t^\top - \boldsymbol{M}^*\right\|_F^2 \leqslant 2\|\boldsymbol{M}^*\|_F^2$ which implies $\left\|\boldsymbol{U}_t\boldsymbol{V}_t^\top\right\|_F \leqslant 3\|\boldsymbol{M}^*\|_F$. Therefore it suffices to prove

$$\left\|\boldsymbol{U}\boldsymbol{V}^\top\right\|_F \leqslant 3\|\boldsymbol{M}^*\|_F, \left\|\boldsymbol{U}^\top\boldsymbol{U} - \boldsymbol{V}^\top\boldsymbol{V}\right\|_F \leqslant \epsilon \implies \|\boldsymbol{U}\|_F^2 \leqslant 5\sqrt{r}\|\boldsymbol{M}^*\|_F, \|\boldsymbol{V}\|_F^2 \leqslant 5\sqrt{r}\|\boldsymbol{M}^*\|_F. \tag{15}$$

Now we prove (15). Consider the SVD $\boldsymbol{U} = \boldsymbol{\Phi}\boldsymbol{\Sigma}\boldsymbol{\Psi}^\top$, where $\boldsymbol{\Phi} \in \mathbb{R}^{d_1 \times d_1}$ and $\boldsymbol{\Psi} \in \mathbb{R}^{r \times r}$ are orthogonal matrices, and $\boldsymbol{\Sigma} \in \mathbb{R}^{d_1 \times r}$ is a diagonal matrix. Let $\sigma_i = \boldsymbol{\Sigma}[i, i]$ ($i \in [r]$) which are all the singular values of $\boldsymbol{U}$. Define $\widetilde{\boldsymbol{V}} = \boldsymbol{V}\boldsymbol{\Psi}$. Then we have

$$3\|\boldsymbol{M}^*\|_F \geqslant \left\|\boldsymbol{U}\boldsymbol{V}^\top\right\|_F = \left\|\boldsymbol{\Phi}\boldsymbol{\Sigma}\boldsymbol{\Psi}^\top\boldsymbol{\Psi}\widetilde{\boldsymbol{V}}^\top\right\|_F = \left\|\boldsymbol{\Sigma}\widetilde{\boldsymbol{V}}^\top\right\|_F = \sqrt{\sum_{i=1}^{r} \sigma_i^2 \left\|\widetilde{\boldsymbol{V}}[:, i]\right\|^2}$$

and

$$\epsilon \geqslant \left\| \boldsymbol{U}^\top \boldsymbol{U} - \boldsymbol{V}^\top \boldsymbol{V} \right\|_F = \left\| \boldsymbol{\Psi} \boldsymbol{\Sigma}^\top \boldsymbol{\Phi}^\top \boldsymbol{\Phi} \boldsymbol{\Sigma} \boldsymbol{\Psi}^\top - \boldsymbol{\Psi} \widetilde{\boldsymbol{V}}^\top \widetilde{\boldsymbol{V}} \boldsymbol{\Psi}^\top \right\|_F = \left\| \boldsymbol{\Sigma}^\top \boldsymbol{\Sigma} - \widetilde{\boldsymbol{V}}^\top \widetilde{\boldsymbol{V}} \right\|_F$$

$$\geqslant \sqrt{\sum_{i=1}^{r} \left( \sigma_i^2 - \left\| \widetilde{\boldsymbol{V}}[:,i] \right\|^2 \right)^2}.$$

Using the above two inequalities we get

$$\sum_{i=1}^{r} \sigma_i^4 \leqslant \sum_{i=1}^{r} \left( \sigma_i^4 + \left\| \widetilde{\boldsymbol{V}}[:,i] \right\|^4 \right) = \sum_{i=1}^{r} \left( \sigma_i^2 - \left\| \widetilde{\boldsymbol{V}}[:,i] \right\|^2 \right)^2 + 2 \sum_{i=1}^{r} \sigma_i^2 \left\| \widetilde{\boldsymbol{V}}[:,i] \right\|^2$$

$$\leqslant \epsilon^2 + 2 \left( 3 \left\| \boldsymbol{M}^* \right\|_F \right)^2 \leqslant 19 \left\| \boldsymbol{M}^* \right\|_F^2.$$

Then by the Cauchy-Schwarz inequality we have

$$\left\| \boldsymbol{U} \right\|_F^2 = \sum_{i=1}^{r} \sigma_i^2 \leqslant \sqrt{r \sum_{i=1}^{r} \sigma_i^4} \leqslant \sqrt{r \cdot 19 \left\| \boldsymbol{M}^* \right\|_F^2} \leqslant 5 \sqrt{r} \left\| \boldsymbol{M}^* \right\|_F.$$

Similarly, we also have $\left\| \boldsymbol{V} \right\|_F^2 \leqslant 5 \sqrt{r} \left\| \boldsymbol{M}^* \right\|_F$. Therefore we have proved (15). $\qquad \square$

## B.2  Convergence to a Stationary Point

With the balancedness and boundedness properties in Lemma 3.1, it is then standard to show that $(\boldsymbol{U}_t, \boldsymbol{V}_t)$ converges to a stationary point of $f$.

**Lemma B.2.** *Under the setting of Theorem 3.1, with high probability* $\lim_{t \to \infty} (\boldsymbol{U}_t, \boldsymbol{V}_t) = (\bar{\boldsymbol{U}}, \bar{\boldsymbol{V}})$ *exists, and* $(\bar{\boldsymbol{U}}, \bar{\boldsymbol{V}})$ *is a stationary point of* $f$. *Furthermore,* $(\bar{\boldsymbol{U}}, \bar{\boldsymbol{V}})$ *satisfies* $\left\| \bar{\boldsymbol{U}}^\top \bar{\boldsymbol{U}} - \bar{\boldsymbol{V}}^\top \bar{\boldsymbol{V}} \right\| \leqslant \epsilon$.

*Proof.* We assume the three properties in Lemma 3.1 hold, which happens with high probability. Then from (14) we have

$$f(\boldsymbol{U}_{t+1}, \boldsymbol{V}_{t+1}) \leqslant f(\boldsymbol{U}_t, \boldsymbol{V}_t) - \frac{\eta_t}{2} \left\| \nabla f(\boldsymbol{U}_t, \boldsymbol{V}_t) \right\|_F^2$$
$$= f(\boldsymbol{U}_t, \boldsymbol{V}_t) - \frac{1}{2} \left\| \nabla f(\boldsymbol{U}_t, \boldsymbol{V}_t) \right\|_F \left\| \begin{pmatrix} \boldsymbol{U}_{t+1} \\ \boldsymbol{V}_{t+1} \end{pmatrix} - \begin{pmatrix} \boldsymbol{U}_t \\ \boldsymbol{V}_t \end{pmatrix} \right\|_F. \tag{16}$$

Under the above descent condition, the result of Absil et al. [2005] says that the iterates either diverge to infinity or converge to a fixed point. According to Lemma 3.1, $\{(\boldsymbol{U}_t, \boldsymbol{V}_t)\}_{t=1}^\infty$ are all bounded, so they have to converge to a fixed point $(\bar{\boldsymbol{U}}, \bar{\boldsymbol{V}})$ as $t \to \infty$.

Next, from (16) we know that $\sum_{t=1}^\infty \frac{\eta_t}{2} \left\| \nabla f(\boldsymbol{U}_t, \boldsymbol{V}_t) \right\|_F^2 \leqslant f(\boldsymbol{U}_0, \boldsymbol{V}_0)$ is bounded. Notice that $\eta_t$ scales like $1/t$. So we must have $\liminf_{t \to \infty} \left\| \nabla f(\boldsymbol{U}_t, \boldsymbol{V}_t) \right\|_F = 0$. Then according to the smoothness of $f$ in a bounded region (Lemma B.1) we conclude $\nabla f(\bar{\boldsymbol{U}}, \bar{\boldsymbol{V}}) = \boldsymbol{0}$, i.e., $(\bar{\boldsymbol{U}}, \bar{\boldsymbol{V}})$ is a stationary point.

The second part of the lemma is evident according to Lemma 3.1 (i). $\qquad \square$

## B.3  Proof of Lemma 3.2

The main idea in the proof is similar to Ge et al. [2017a]. We want to find a direction $\boldsymbol{\Delta}$ such that either $[\nabla^2 f(\boldsymbol{U}, \boldsymbol{V})](\boldsymbol{\Delta}, \boldsymbol{\Delta})$ is negative or $(\boldsymbol{U}, \boldsymbol{V})$ is close to a global minimum. We show that this is possible when $\left\| \boldsymbol{U}^\top \boldsymbol{U} - \boldsymbol{V}^\top \boldsymbol{V} \right\|_F \leqslant \epsilon$.

First we define some notation. Take the SVD $\boldsymbol{M}^* = \boldsymbol{\Phi}^* \boldsymbol{\Sigma}^* \boldsymbol{\Psi}^{*\top}$, where $\boldsymbol{\Phi}^* \in \mathbb{R}^{d_1 \times r}$ and $\boldsymbol{\Psi}^* \in \mathbb{R}^{d_2 \times r}$ have orthonormal columns and $\boldsymbol{\Sigma}^* \in \mathbb{R}^{r \times r}$ is diagonal. Denote $\boldsymbol{U}^* = \boldsymbol{\Phi}^* (\boldsymbol{\Sigma}^*)^{1/2}$ and $\boldsymbol{V}^* = \boldsymbol{\Psi}^* (\boldsymbol{\Sigma}^*)^{1/2}$. Then we have $\boldsymbol{U}^* \boldsymbol{V}^{*\top} = \boldsymbol{M}^*$ (i.e., $(\boldsymbol{U}^*, \boldsymbol{V}^*)$ is a global minimum) and $\boldsymbol{U}^{*\top} \boldsymbol{U}^* = \boldsymbol{V}^{*\top} \boldsymbol{V}^*$.

Let $\boldsymbol{M} = \boldsymbol{U} \boldsymbol{V}^\top$, $\boldsymbol{W} = \begin{pmatrix} \boldsymbol{U} \\ \boldsymbol{V} \end{pmatrix}$ and $\boldsymbol{W}^* = \begin{pmatrix} \boldsymbol{U}^* \\ \boldsymbol{V}^* \end{pmatrix}$. Define

$$\boldsymbol{R} = \operatorname{argmin}_{\boldsymbol{R}' \in \mathbb{R}^{r \times r}, \text{ orthogonal}} \left\| \boldsymbol{W} - \boldsymbol{W}^* \boldsymbol{R}' \right\|_F$$

and
$$\boldsymbol{\Delta} = \boldsymbol{W} - \boldsymbol{W}^* \boldsymbol{R}.$$

We will show that $\boldsymbol{\Delta}$ is the desired direction. Recall (12):

$$[\nabla^2 f(\boldsymbol{U}, \boldsymbol{V})](\boldsymbol{\Delta}, \boldsymbol{\Delta}) = 2\langle \boldsymbol{M} - \boldsymbol{M}^*, \boldsymbol{\Delta}_U \boldsymbol{\Delta}_V^\top \rangle + \|\boldsymbol{U}\boldsymbol{\Delta}_V^\top + \boldsymbol{\Delta}_U \boldsymbol{V}^\top\|_F^2, \qquad (17)$$

where $\boldsymbol{\Delta} = \begin{pmatrix} \boldsymbol{\Delta}_U \\ \boldsymbol{\Delta}_V \end{pmatrix}$, $\boldsymbol{\Delta}_U \in \mathbb{R}^{d_1 \times r}$, $\boldsymbol{\Delta}_V \in \mathbb{R}^{d_2 \times r}$. We consider the two terms in (17) separately.

For the first term in (17), we have:

**Claim B.4.** $\langle \boldsymbol{M} - \boldsymbol{M}^*, \boldsymbol{\Delta}_U \boldsymbol{\Delta}_V^\top \rangle = -\|\boldsymbol{M} - \boldsymbol{M}^*\|_F^2.$

*Proof.* Since $(\boldsymbol{U}, \boldsymbol{V})$ is a stationary point of $f$, we have the first-order optimality condition:

$$\frac{\partial f(\boldsymbol{U}, \boldsymbol{V})}{\partial \boldsymbol{U}} = (\boldsymbol{M} - \boldsymbol{M}^*)\boldsymbol{V} = \boldsymbol{0}, \qquad \frac{\partial f(\boldsymbol{U}, \boldsymbol{V})}{\partial \boldsymbol{V}} = (\boldsymbol{M} - \boldsymbol{M}^*)^\top \boldsymbol{U} = \boldsymbol{0}. \qquad (18)$$

Note that $\boldsymbol{\Delta}_U = \boldsymbol{U} - \boldsymbol{U}^*\boldsymbol{R}$ and $\boldsymbol{\Delta}_V = \boldsymbol{V} - \boldsymbol{V}^*\boldsymbol{R}$. We have

$$\begin{aligned}
&\langle \boldsymbol{M} - \boldsymbol{M}^*, \boldsymbol{\Delta}_U \boldsymbol{\Delta}_V^\top \rangle \\
&= \langle \boldsymbol{M} - \boldsymbol{M}^*, (\boldsymbol{U} - \boldsymbol{U}^*\boldsymbol{R})(\boldsymbol{V} - \boldsymbol{V}^*\boldsymbol{R})^\top \rangle \\
&= \langle \boldsymbol{M} - \boldsymbol{M}^*, \boldsymbol{M} - \boldsymbol{U}^*\boldsymbol{R}\boldsymbol{V}^\top - \boldsymbol{U}\boldsymbol{R}^\top\boldsymbol{V}^{*\top} + \boldsymbol{M}^* \rangle \\
&= \langle \boldsymbol{M} - \boldsymbol{M}^*, \boldsymbol{M}^* \rangle \\
&= \langle \boldsymbol{M} - \boldsymbol{M}^*, \boldsymbol{M}^* - \boldsymbol{M} \rangle \\
&= -\|\boldsymbol{M} - \boldsymbol{M}^*\|_F^2,
\end{aligned}$$

where we have used the following consequences of (18):

$$\begin{aligned}
\langle \boldsymbol{M} - \boldsymbol{M}^*, \boldsymbol{M} \rangle &= \langle \boldsymbol{M} - \boldsymbol{M}^*, \boldsymbol{U}\boldsymbol{V}^\top \rangle = 0, \\
\langle \boldsymbol{M} - \boldsymbol{M}^*, \boldsymbol{U}^*\boldsymbol{R}\boldsymbol{V}^\top \rangle &= 0, \\
\langle \boldsymbol{M} - \boldsymbol{M}^*, \boldsymbol{U}\boldsymbol{R}^\top\boldsymbol{V}^{*\top} \rangle &= 0.
\end{aligned}$$

$\square$

The second term in (17) has the following upper bound:

**Claim B.5.** $\|\boldsymbol{U}\boldsymbol{\Delta}_V + \boldsymbol{\Delta}_U\boldsymbol{V}\|_F^2 \leqslant \|\boldsymbol{M} - \boldsymbol{M}^*\|_F^2 + \frac{1}{2}\epsilon^2.$

*Proof.* We make use of the following identities, all of which can be directly verified by plugging in definitions:

$$\boldsymbol{U}\boldsymbol{\Delta}_V^\top + \boldsymbol{\Delta}_U\boldsymbol{V}^\top = \boldsymbol{\Delta}_U\boldsymbol{\Delta}_V^\top + \boldsymbol{M} - \boldsymbol{M}^*, \qquad (19)$$

$$\|\boldsymbol{\Delta}\boldsymbol{\Delta}^\top\|_F^2 = 4\|\boldsymbol{\Delta}_U\boldsymbol{\Delta}_V^\top\|_F^2 + \|\boldsymbol{\Delta}_U^\top\boldsymbol{\Delta}_U - \boldsymbol{\Delta}_V^\top\boldsymbol{\Delta}_V\|_F^2, \qquad (20)$$

$$\begin{aligned}
\|\boldsymbol{W}\boldsymbol{W}^\top - \boldsymbol{W}^*\boldsymbol{W}^{*\top}\|_F^2 = 4\|\boldsymbol{M} - \boldsymbol{M}^*\|_F^2 - 2\|\boldsymbol{U}^\top\boldsymbol{U}^* - \boldsymbol{V}^\top\boldsymbol{V}^*\|_F^2 \\
+ \|\boldsymbol{U}^\top\boldsymbol{U} - \boldsymbol{V}^\top\boldsymbol{V}\|_F^2 + \|\boldsymbol{U}^{*\top}\boldsymbol{U}^* - \boldsymbol{V}^{*\top}\boldsymbol{V}^*\|_F^2.
\end{aligned} \qquad (21)$$

We also need the following inequality, which is [Ge et al., 2017a, Lemma 6]:

$$\|\boldsymbol{\Delta}\boldsymbol{\Delta}^\top\|_F^2 \leqslant 2\|\boldsymbol{W}\boldsymbol{W}^\top - \boldsymbol{W}^*\boldsymbol{W}^{*\top}\|_F^2. \qquad (22)$$

Now we can prove the desired bound as follows:

$$\begin{aligned}
&\|\boldsymbol{U}\boldsymbol{\Delta}_V + \boldsymbol{\Delta}_U\boldsymbol{V}\|_F^2 \\
&= \|\boldsymbol{\Delta}_U\boldsymbol{\Delta}_V^\top + \boldsymbol{M} - \boldsymbol{M}^*\|_F^2 && ((19)) \\
&= \|\boldsymbol{\Delta}_U\boldsymbol{\Delta}_V^\top\|_F^2 + 2\langle \boldsymbol{M} - \boldsymbol{M}^*, \boldsymbol{\Delta}_U\boldsymbol{\Delta}_V^\top \rangle + \|\boldsymbol{M} - \boldsymbol{M}^*\|_F^2
\end{aligned}$$

$$= \left\| \boldsymbol{\Delta}_U \boldsymbol{\Delta}_V^\top \right\|_F^2 - \left\| \boldsymbol{M} - \boldsymbol{M}^* \right\|_F^2 \qquad \text{(Claim B.4)}$$

$$\leqslant \frac{1}{4} \left\| \boldsymbol{\Delta}\boldsymbol{\Delta}^\top \right\|_F^2 - \left\| \boldsymbol{M} - \boldsymbol{M}^* \right\|_F^2 \qquad ((20))$$

$$\leqslant \frac{1}{2} \left\| \boldsymbol{W}\boldsymbol{W}^\top - \boldsymbol{W}^*\boldsymbol{W}^{*\top} \right\|_F^2 - \left\| \boldsymbol{M} - \boldsymbol{M}^* \right\|_F^2 \qquad ((22))$$

$$= 2 \left\| \boldsymbol{M} - \boldsymbol{M}^* \right\|_F^2 - \left\| \boldsymbol{U}^\top \boldsymbol{U}^* - \boldsymbol{V}^\top \boldsymbol{V}^* \right\|_F^2 + \frac{1}{2} \left\| \boldsymbol{U}^\top \boldsymbol{U} - \boldsymbol{V}^\top \boldsymbol{V} \right\|_F^2$$

$$+ \frac{1}{2} \left\| \boldsymbol{U}^{*\top} \boldsymbol{U}^* - \boldsymbol{V}^{*\top} \boldsymbol{V}^* \right\|_F^2 - \left\| \boldsymbol{M} - \boldsymbol{M}^* \right\|_F^2 \qquad ((21))$$

$$\leqslant \left\| \boldsymbol{M} - \boldsymbol{M}^* \right\|_F^2 + \frac{1}{2}\epsilon^2,$$

where in the last line we have used $\boldsymbol{U}^{*\top}\boldsymbol{U}^* = \boldsymbol{V}^{*\top}\boldsymbol{V}^*$ and $\left\| \boldsymbol{U}^\top\boldsymbol{U} - \boldsymbol{V}^\top\boldsymbol{V} \right\| \leqslant \epsilon$. $\qquad \square$

Using Claims B.4 and B.5, we obtain an upper bound on (17):

$$[\nabla^2 f(\boldsymbol{U}, \boldsymbol{V})](\boldsymbol{\Delta}, \boldsymbol{\Delta}) \leqslant - \left\| \boldsymbol{M} - \boldsymbol{M}^* \right\|_F^2 + \frac{1}{2}\epsilon^2.$$

Therefore, we have either $\left\| \boldsymbol{U}\boldsymbol{V}^\top - \boldsymbol{M}^* \right\|_F = \left\| \boldsymbol{M} - \boldsymbol{M}^* \right\|_F \leqslant \epsilon$ or $[\nabla^2 f(\boldsymbol{U}, \boldsymbol{V})](\boldsymbol{\Delta}, \boldsymbol{\Delta}) \leqslant -\frac{1}{2}\epsilon^2 < 0$. In the latter case, $(\boldsymbol{U}, \boldsymbol{V})$ is a strict saddle point of $f$. This completes the proof of Lemma 3.2.

### B.4 Finishing the Proof of Theorem 3.1

Theorem 3.1 is a direct corollary of Lemma B.2, Lemma 3.2, and the fact that gradient descent does not converge to a strict saddle point almost surely [Lee et al., 2016, Panageas and Piliouras, 2016].

## C Proof for Rank-1 Matrix Factorization (Theorem 3.2)

In this section we prove Theorem 3.2.

*Proof of Theorem 3.2.* We define the following four key quantities:

$$\alpha_t = \boldsymbol{u}_t^\top \boldsymbol{u}^*, \qquad \alpha_{t,\perp} = \left\| \boldsymbol{U}_\perp^* \boldsymbol{u}_t \right\|_2, \qquad \beta_t = \boldsymbol{v}_t^\top \boldsymbol{v}^*, \qquad \beta_{t,\perp} = \left\| \boldsymbol{V}_\perp^* \boldsymbol{v}_t \right\|_2,$$

where $\boldsymbol{U}_\perp^* = \boldsymbol{I} - \boldsymbol{u}^* \boldsymbol{u}^{*\top}$ and $\boldsymbol{V}_\perp^* = \boldsymbol{I} - \boldsymbol{v}^* \boldsymbol{v}^{*\top}$ are the projection matrices onto the orthogonal complement spaces of $\boldsymbol{u}^*$ and $\boldsymbol{v}^*$, respectively. Notice that $\|\boldsymbol{u}_t\|_2^2 = \alpha_t^2 + \alpha_{t,\perp}^2$ and $\|\boldsymbol{v}_t\|_2^2 = \beta_t^2 + \beta_{t,\perp}^2$. It turns out that we can write down the explicit formulas for the dynamics of these quantities:

$$\alpha_{t+1} = \left(1 - \eta\left(\beta_t^2 + \beta_{t,\perp}^2\right)\right)\alpha_t + \eta\sigma_1\beta_t, \qquad \beta_{t+1} = \left(1 - \eta\left(\alpha_t^2 + \alpha_{t,\perp}^2\right)\right)\beta_t + \eta_1\sigma_1\alpha_t,$$
$$\alpha_{t+1,\perp} = \left(1 - \eta\left(\beta_t^2 + \beta_{t,\perp}^2\right)\right)\alpha_{t,\perp}, \qquad \beta_{t+1,\perp} = \left(1 - \eta\left(\alpha_t^2 + \alpha_{t,\perp}^2\right)\right)\beta_{t,\perp}. \tag{23}$$

To facilitate the analysis, we also define:

$$h_t = \alpha_t\beta_t - \sigma_1,$$
$$\xi_t = \alpha_{t,\perp}^2 + \beta_{t,\perp}^2.$$

Then our goal is to show $\xi_t \to 0$ and $h_t \to 0$ as $t \to \infty$. We calculate the dynamics of $h_t$ and $\xi_t$:

$$h_{t+1} = \left(1 - \eta\left(\alpha_t^2 + \beta_t^2\right) + \eta^2\left(\alpha_t\beta_t h_t + \alpha_t^2\beta_{t,\perp}^2 + \beta_t^2\alpha_{t,\perp}^2 + \alpha_{t,\perp}^2\beta_{t,\perp}^2\right)\right)h_t - \eta\alpha_t\beta_t\xi_t + \eta^2\sigma_1\alpha_{t,\perp}^2\beta_{t,\perp}^2,$$
$$\xi_{t+1} = \left(1 - \eta\left(\beta_t^2 + \beta_{t,\perp}^2\right)\right)^2\alpha_{t,\perp}^2 + \left(1 - \eta\left(\alpha_t^2 + \alpha_{t,\perp}^2\right)\right)^2\beta_{t,\perp}^2.$$
$$\tag{24}$$

According to our initialization scheme, with high probability we have $|\alpha_0|, |\beta_0| \in \left[0.1 c_{init}\sqrt{\frac{\sigma_1}{d}}, 10 c_{init}\sqrt{\frac{\sigma_1}{d}}\right]$ and $|\alpha_{0,\perp}|, |\beta_{0,\perp}| \leqslant 10 c_{init}\sqrt{\sigma_1}$. We assume that these conditions are

satisfied. We also assume that the signal at the beginning is positive: $\alpha_0 \beta_0 > 0$, which holds with probability $1/2$. Without loss of generality we assume $\alpha_0, \beta_0 > 0$.[12]

We divide the dynamics into two stages.

**Lemma C.1** (Stage 1: escaping from saddle point $(\mathbf{0}, \mathbf{0})$). *Let* $T_1 = \min\left\{t \in \mathbb{N} : \alpha_t^2 + \beta_t^2 \geqslant \frac{1}{2}\sigma_1\right\}$. *Then for* $t = 0, 1, \ldots, T_1 - 1$, *the followings hold:*

(i) *Positive signal strengths:* $\alpha_t, \beta_t > 0$;

(ii) *Small magnitudes in complement space:* $\xi_t \leqslant \xi_0 \leqslant 100 c_{init}^2 \sigma_1$;

(iii) *Growth of magnitude in signal space:* $\left(1 + \frac{c_{step}}{3}\right)(\alpha_t + \beta_t) \leqslant \alpha_{t+1} + \beta_{t+1} \leqslant (1 + c_{step})(\alpha_t + \beta_t)$;

(iv) *Bounded ratio between two layers:* $|\alpha_t - \beta_t| \leqslant \frac{99}{101}(\alpha_t + \beta_t)$.

*Furthermore, we have* $T_1 = O(\log d)$.

In this stage, the strengths in the complement spaces remain small ($\xi_t \leqslant \xi_0$) and the strength in the signal space is growing exponentially ($\alpha_{t+1} + \beta_{t+1} \geqslant \left(1 + \frac{c_{step}}{3}\right)(\alpha_t + \beta_t)$). Furthermore, $|\alpha_t - \beta_t| \leqslant \frac{99}{101}(\alpha_t + \beta_t)$ implies $\frac{\alpha_t}{\beta_t} \in \left[\frac{1}{100}, 100\right]$, which means the signal strengths in the two layers are of the same order.

Then we enter stage 2, which is essentially a local convergence phase. The following lemma characterizes the behaviors of the strengths in the signal and noise spaces in this stage.

**Lemma C.2** (Stage 2: convergence to global minimum). *Let* $T_1$ *be as defined in Lemma C.1. Then there exists a universal constant* $c_1 > 0$ *such that the followings hold for all* $t \geqslant T_1$:

(a) *Non-vanishing signal strengths in both layers:* $\alpha_t, \beta_t \geqslant \sqrt{c_1 \sigma_1}$;

(b) *Bounded signal strengths:* $\alpha_t \beta_t \leqslant \sigma_1$, *i.e.,* $h_t \leqslant 0$;

(c) *Shrinking magnitudes in complement spaces:* $\xi_t \leqslant (1 - c_1 c_{step})^{t-T_1} \xi_0 \leqslant (1 - c_1 c_{step})^{t-T_1} \cdot 100 c_{init}^2 \sigma_1$;

(d) *Convergence in signal space:* $|h_{t+1}| \leqslant (1 - c_1 c_{step})|h_t| + c_{step}\xi_t$.

Note that properties (a) and (b) in Lemma C.2 imply $c_0 \leqslant \frac{\alpha_t}{\beta_t} \leqslant C_0$ for all $t \geqslant T_1$, where $c_0, C_0 > 0$ are universal constants. Property (c) implies that for all $t \geqslant T_1 + T_2$ where $T_2 = \Theta(\log\frac{1}{\epsilon})$, we have $\xi_t = O(\epsilon\sigma_1)$. Then property (d) tells us that after another $T_3 = \Theta(\log\frac{1}{\epsilon})$ iterations, we can ensure $|h_t| = O(\epsilon\sigma_1)$ for all $t \geqslant T_1 + T_2 + T_3$. These imply $\left\| \boldsymbol{u}_t \boldsymbol{v}_t^\top - \boldsymbol{M}^* \right\|_F = O(\epsilon\sigma_1)$ after $t = T_1 + T_2 + T_3 = O(\log\frac{d}{\epsilon})$ iterations, completing the proof of Theorem 3.2. $\square$

Now we prove Lemmas C.1 and C.2.

*Proof of Lemma C.1.* We use induction to prove the following statements for $t = 0, 1, \ldots, T_1 - 1$:

$$
\begin{aligned}
\mathcal{D}(t): \quad & \alpha_t, \beta_t > 0, \\
\mathcal{E}(t): \quad & \xi_t \leqslant \xi_0 \leqslant 100 c_{init}^2 \sigma_1, \\
\mathcal{F}(t): \quad & \left(1 + \frac{c_{step}}{3}\right)(\alpha_t + \beta_t) \leqslant \alpha_{t+1} + \beta_{t+1} \leqslant (1 + c_{step})(\alpha_t + \beta_t), \\
\mathcal{G}(t): \quad & |\alpha_t - \beta_t| \leqslant \frac{99}{101}(\alpha_t + \beta_t), \\
\mathcal{H}(t): \quad & \|\boldsymbol{u}_t\|^2 + \|\boldsymbol{v}_t\|^2 \leqslant \sigma_1.
\end{aligned}
$$

- Base cases.

  We know that $\mathcal{D}(0)$, $\mathcal{E}(0)$ and $\mathcal{G}(0)$ hold from our assumptions on the initialization.

- $\mathcal{D}(t), \mathcal{E}(t) \implies \mathcal{F}(t)$ $(\forall t \leqslant T_1 - 1)$.

  From (23) we have

$$
\begin{aligned}
\alpha_{t+1} + \beta_{t+1} &= (1 + \eta\sigma_1)(\alpha_t + \beta_t) - \eta\left(\alpha_t^2 + \alpha_{t,\perp}^2\right)\beta_t - \eta\left(\beta_t^2 + \beta_{t,\perp}^2\right)\alpha_t \\
&\geqslant \left(1 + \eta\sigma_1 - \eta\left(\alpha_t^2 + \beta_t^2 + \xi_t\right)\right)(\alpha_t + \beta_t) \\
&\geqslant \left(1 + \eta\sigma_1 - \eta\left(\frac{\sigma_1}{2} + 100c_{init}^2\sigma_1\right)\right)(\alpha_t + \beta_t) \\
&\geqslant \left(1 + \frac{\eta\sigma_1}{3}\right)(\alpha_t + \beta_t) \\
&= \left(1 + \frac{c_{step}}{3}\right)(\alpha_t + \beta_t),
\end{aligned}
$$

  where in the second inequality we have used the definition of $T_1$, and the last inequality is true when $c_{init}$ is sufficiently small.

  On the other hand we have

$$
\begin{aligned}
\alpha_{t+1} + \beta_{t+1} &= (1 + \eta\sigma_1)(\alpha_t + \beta_t) - \eta\left(\alpha_t^2 + \alpha_{t,\perp}^2\right)\beta_t - \eta\left(\beta_t^2 + \beta_{t,\perp}^2\right)\alpha_t \\
&\leqslant (1 + \eta\sigma_1)(\alpha_t + \beta_t) \\
&= (1 + c_{step})(\alpha_t + \beta_t).
\end{aligned}
$$

- $\mathcal{E}(t) \implies \mathcal{H}(t)$ $(\forall t \leqslant T_1 - 1)$.

  We have

$$
\|\boldsymbol{u}_t\|^2 + \|\boldsymbol{v}_t\|^2 = \alpha_t^2 + \beta_t^2 + \xi_t \leqslant \frac{1}{2}\sigma_1 + 100c_{init}^2\sigma_1 \leqslant \sigma_1.
$$

- $\mathcal{D}(t), \mathcal{H}(t) \implies \mathcal{D}(t+1)$ $(\forall t \leqslant T_1 - 1)$.

  From (23) we have

$$
\alpha_{t+1} = \left(1 - \eta\|\boldsymbol{v}_t\|^2\right)\alpha_t + \eta\sigma_1\beta_t \geqslant (1 - \eta\sigma_1)\alpha_t = (1 - c_{step})\alpha_t > 0.
$$

  Similarly we have $\beta_{t+1} > 0$. Note that $c_{step}$ is chosen to be sufficiently small.

- $\mathcal{H}(t) \implies \mathcal{E}(t+1)$ $(\forall t \leqslant T_1 - 1)$.

  Recall from (24):

$$
\xi_{t+1} = \left(1 - \eta\|\boldsymbol{v}_t\|^2\right)^2\alpha_{t,\perp}^2 + \left(1 - \eta\|\boldsymbol{u}_t\|^2\right)^2\beta_{t,\perp}^2.
$$

  Since $\eta\|\boldsymbol{v}_t\|^2 \leqslant \eta(\|\boldsymbol{u}_t\|^2 + \|\boldsymbol{v}_t\|^2) \leqslant \eta\sigma_1 = c_{step} \leqslant 1$ and $\eta\|\boldsymbol{u}_t\|^2 \leqslant 1$, we have

$$
\xi_{t+1} \leqslant \alpha_{t,\perp}^2 + \beta_{t,\perp}^2 = \xi_t.
$$

- $\mathcal{D}(t), \mathcal{E}(t), \mathcal{F}(t), \mathcal{G}(t) \implies \mathcal{G}(t+1)$ $(\forall t \leqslant T_1 - 1)$.

  From (23) we have

$$
\begin{aligned}
\alpha_{t+1} - \beta_{t+1} &= (1 - \eta\sigma_1)(\alpha_t - \beta_t) - \eta(\beta_t^2 + \beta_{t,\perp}^2)\alpha_t + \eta(\alpha_t^2 + \alpha_{t,\perp}^2)\beta_t \\
&= (1 - \eta\sigma_1 + \eta\alpha_t\beta_t)(\alpha_t - \beta_t) - \eta\beta_{t,\perp}^2\alpha_t + \eta\alpha_{t,\perp}^2\beta_t.
\end{aligned}
$$

  From $\alpha_t^2 + \beta_t^2 < \frac{1}{2}\sigma_1$ we know $\alpha_t\beta_t < \frac{1}{4}\sigma_1$. Thus

$$
\begin{aligned}
|\alpha_{t+1} - \beta_{t+1}| &\leqslant (1 - \eta\sigma_1 + \eta\alpha_t\beta_t)\,|\alpha_t - \beta_t| + \eta\beta_{t,\perp}^2\alpha_t + \eta\alpha_{t,\perp}^2\beta_t \\
&\leqslant \left(1 - \frac{3}{4}\eta\sigma_1\right)|\alpha_t - \beta_t| + \eta\xi_t(\alpha_t + \beta_t)
\end{aligned}
$$

$$\leqslant \left(1 - \frac{3}{4}\eta\sigma_1\right) \cdot \frac{99}{101}(\alpha_t + \beta_t) + \eta \cdot 100c_{init}^2\sigma_1(\alpha_t + \beta_t)$$

$$\leqslant \left(1 - \eta\sigma_1\left(\frac{3}{4} - 100c_{init}^2 \cdot \frac{101}{99}\right)\right) \cdot \frac{99}{101}(\alpha_t + \beta_t)$$

$$\leqslant \frac{99}{101}(\alpha_t + \beta_t)$$

$$\leqslant \frac{99}{101}(\alpha_{t+1} + \beta_{t+1}).$$

Lastly we upper bound $T_1$. Note that for all $t < T_1$ we have $\alpha_t + \beta_t \leqslant \sqrt{2(\alpha_t^2 + \beta_t^2)} < \sqrt{2 \cdot \frac{1}{2}\sigma_1} = \sqrt{\sigma_1}$. From $\mathcal{F}(t)$ we know that $\alpha_t + \beta_t$ is increasing exponentially. Therefore, we must have $T_1 = O\left(\log \frac{\sqrt{\sigma_1}}{\alpha_0 + \beta_0}\right) = O\left(\log \frac{\sqrt{\sigma_1}}{\sqrt{\sigma_1/d}}\right) = O(\log d)$. $\qquad\square$

*Proof of Lemma C.2.* By the definition of $T_1$ we know $\alpha_{T_1}^2 + \beta_{T_1}^2 \geqslant \frac{1}{2}\sigma_1$. In the proof of Lemma C.1, we have shown $\alpha_{T_1}, \beta_{T_1} > 0$ and $|\alpha_{T_1} - \beta_{T_1}| \leqslant \frac{99}{101}(\alpha_{T_1} + \beta_{T_1})$. These imply $\min\{\alpha_{T_1}, \beta_{T_1}\} \geqslant 2\sqrt{c_1\sigma_1}$ for some small universal constant $c_1 > 0$.

We use induction to prove the following statements for all $t \geqslant T_1$:

$$\mathcal{I}(t): \quad \alpha_t \geqslant \alpha_{T_1} \cdot \prod_{i=T_1}^{t-1}\left(1 - \eta\xi_0\left(1 - c_1c_{step}\right)^{i-T_1}\right), \quad \beta_t \geqslant \beta_{T_1} \cdot \prod_{i=T_1}^{t-1}\left(1 - \eta\xi_0\left(1 - c_1c_{step}\right)^{i-T_1}\right),$$

$$\mathcal{J}(t): \quad \alpha_t, \beta_t \geqslant \sqrt{c_1\sigma_1},$$

$$\mathcal{K}(t): \quad \alpha_t\beta_t \leqslant \sigma_1, \text{ i.e., } h_t \leqslant 0,$$

$$\mathcal{L}(t): \quad \xi_t \leqslant (1 - c_1c_{step})^{t-T_1}\xi_0 \leqslant (1 - c_1c_{step})^{t-T_1} \cdot 100c_{init}^2\sigma_1,$$

$$\mathcal{M}(t): \quad |h_{t+1}| \leqslant (1 - c_1c_{step})|h_t| + c_{step}\xi_t.$$

- Base cases.

  $\mathcal{I}(T_1)$ is obvious. We know that $\mathcal{J}(T_1)$ is true by the definition of $c_1$. $\mathcal{K}(T_1)$ can be shown as follows:

$$\alpha_{T_1}\beta_{T_1} \leqslant \frac{1}{4}(\alpha_{T_1} + \beta_{T_1})^2$$

$$\leqslant \frac{1}{4}(1 + c_{step})^2(\alpha_{T_1-1} + \beta_{T_1-1})^2 \qquad \text{(by Lemma C.1 (iii))}$$

$$\leqslant \frac{1}{4}(1 + c_{step})^2 \cdot 2\left(\alpha_{T_1-1}^2 + \beta_{T_1-1}^2\right)$$

$$\leqslant \frac{1}{4}(1 + c_{step})^2 \cdot 2 \cdot \frac{1}{2}\sigma_1 \qquad \text{(by the definition of } T_1\text{)}$$

$$\leqslant \sigma_1. \qquad \text{(choosing } c_{step} \text{ to be small)}$$

  $\mathcal{L}(T_1)$ reduces to $\xi_{T_1} \leqslant \xi_0$, which was shown in the proof of Lemma C.1.

- $\mathcal{I}(t) \implies \mathcal{J}(t)$ $(\forall t \geqslant T_1)$.

  Notice that we have $\eta\xi_0 \leqslant \frac{c_{step}}{\sigma_1} \cdot 100c_{init}^2\sigma_1 = 100c_{step}c_{init}^2 < \frac{1}{2}$ since $c_{step}$ and $c_{init}$ are sufficiently small. Then we have

$$\alpha_t \geqslant \alpha_{T_1} \cdot \prod_{i=T_1}^{t-1}\left(1 - \eta\xi_0\left(1 - c_1c_{step}\right)^{i-T_1}\right)$$

$$\geqslant \alpha_{T_1} \cdot \prod_{i=0}^{\infty}\left(1 - \eta\xi_0\left(1 - c_1c_{step}\right)^i\right)$$

$$\geqslant \alpha_{T_1} \cdot \prod_{i=0}^{\infty}\exp\left(-2\eta\xi_0\left(1 - c_1c_{step}\right)^i\right) \qquad (1 - x \geqslant e^{-2x}, \forall 0 \leqslant x \leqslant 1/2)$$

$$= \alpha_{T_1} \cdot \exp\left(-\frac{2\eta\xi_0}{c_1 c_{step}}\right)$$

$$\geqslant \alpha_{T_1} \cdot \exp\left(-\frac{200 c_{step} c_{init}^2}{c_1 c_{step}}\right)$$

$$\geqslant 2\sqrt{c_1 \sigma_1} \cdot \exp\left(-\frac{200 c_{init}^2}{c_1}\right)$$

$$\geqslant \sqrt{c_1 \sigma_1}. \qquad\qquad \text{(choosing } c_{init} \text{ to be small)}$$

Similarly we have $\beta_t \geqslant \sqrt{c_1 \sigma_1}$.

- $\mathcal{I}(t), \mathcal{J}(t), \mathcal{K}(t), \mathcal{L}(t) \implies \mathcal{I}(t+1)$ ($\forall t \geqslant T_1$).

  From (23) we have

$$\alpha_{t+1} = \left(1 - \eta\left(\beta_t^2 + \beta_{t,\perp}^2\right)\right)\alpha_t + \eta\sigma_1\beta_t$$

$$= \left(1 - \eta\beta_{t,\perp}^2\right)\alpha_t - \eta h_t \beta_t$$

$$\geqslant \left(1 - \eta\beta_{t,\perp}^2\right)\alpha_t \qquad\qquad (h_t \leqslant 0, \beta_t > 0)$$

$$\geqslant \left(1 - \eta\xi_t\right)\alpha_t$$

$$\geqslant \left(1 - \eta\xi_0(1 - c_1 c_{step})^{t-T_1}\right)\alpha_t \qquad\qquad (\mathcal{L}(t))$$

$$\geqslant \alpha_{T_1} \cdot \prod_{i=T_1}^{t}\left(1 - \eta\xi_0\left(1 - c_1 c_{step}\right)^{i-T_1}\right). \qquad\qquad (\mathcal{I}(t))$$

Similarly we have $\beta_{t+1} \geqslant \beta_{T_1} \cdot \prod_{i=T_1}^{t}\left(1 - \eta\xi_0\left(1 - c_1 c_{step}\right)^{i-T_1}\right)$.

- $\mathcal{J}(t), \mathcal{K}(t), \mathcal{L}(t) \implies \mathcal{K}(t+1)$ ($\forall t \geqslant T_1$).

  From (24) we have

$$h_{t+1} = \left(1 - \eta\left(\alpha_t^2 + \beta_t^2\right) + \eta^2\left(\alpha_t\beta_t h_t + \alpha_t^2\beta_{t,\perp}^2 + \beta_t^2\alpha_{t,\perp}^2 + \alpha_{t,\perp}^2\beta_{t,\perp}^2\right)\right)h_t - \eta\alpha_t\beta_t\xi_t + \eta^2\sigma_1\alpha_{t,\perp}^2\beta_{t,\perp}^2$$

$$\leqslant \left(1 - \eta\left(\alpha_t^2 + \beta_t^2\right)\right)h_t + \eta^2\alpha_t\beta_t h_t^2 - \eta\alpha_t\beta_t\xi_t + \eta^2\sigma_1\alpha_{t,\perp}^2\beta_{t,\perp}^2,$$

(25)

where we have used $h_t \leqslant 0$. Since $\alpha_t, \beta_t \geqslant \sqrt{c_1 \sigma_1}$ and $\alpha_t\beta_t \leqslant \sigma_1$, we have $\alpha_t, \beta_t = \Theta(\sqrt{\sigma_1})$. Furthermore, we can choose $c_{step}$ and $c_{init}$ small enough such that $\eta\xi_0 \leqslant 4c_1$ which implies

$$\eta^2\sigma_1\alpha_{t,\perp}^2\beta_{t,\perp}^2 \leqslant \eta^2\sigma_1 \cdot \frac{1}{4}\xi_t^2 \leqslant \frac{1}{4}\eta\xi_t \cdot \eta\sigma_1\xi_0 \leqslant \eta\xi_t \cdot c_1\sigma_1 \leqslant \eta\xi_t \cdot \alpha_t\beta_t.$$

Therefore (25) implies

$$h_{t+1} \leqslant \left(1 - \eta \cdot O(\sigma_1)\right)h_t + \eta^2\alpha_t\beta_t h_t^2$$

$$= \left(1 - \eta \cdot O(\sigma_1) + \eta^2\alpha_t\beta_t h_t\right)h_t$$

$$\leqslant \left(1 - \eta \cdot O(\sigma_1) - \eta^2\sigma_1^2\right)h_t \qquad\qquad (0 < \alpha_t\beta_t \leqslant \sigma_1)$$

$$= \left(1 - O(c_{step}) - c_{step}^2\right)h_t$$

$$\leqslant 0,$$

where the last step is true when $c_{step}$ is sufficiently small.

- $\mathcal{J}(t), \mathcal{K}(t), \mathcal{L}(t) \implies \mathcal{L}(t+1)$ ($\forall t \geqslant T_1$).

  From $\alpha_t, \beta_t \geqslant \sqrt{c_1 \sigma_1}$ and $\alpha_t\beta_t \leqslant \sigma_1$ we have $\alpha_t, \beta_t = \Theta(\sqrt{\sigma_1})$. Also we have $\xi_t \leqslant \xi_0$. Thus we can make sure $\eta(\alpha_t^2 + \alpha_{t,\perp}^2) < 1$ and $\eta(\beta_t^2 + \beta_{t,\perp}^2) < 1$. Then from (24) we have

$$\xi_{t+1} = \left(1 - \eta\left(\beta_t^2 + \beta_{t,\perp}^2\right)\right)^2\alpha_{t,\perp}^2 + \left(1 - \eta\left(\alpha_t^2 + \alpha_{t,\perp}^2\right)\right)^2\beta_{t,\perp}^2$$

$$\leqslant \left(1 - \eta\beta_t^2\right)^2\alpha_{t,\perp}^2 + \left(1 - \eta\alpha_t^2\right)\beta_{t,\perp}^2$$

$$\leqslant \left(1 - \eta c_1\sigma_1\right)\xi_t$$

$$= \left(1 - c_1 c_{step}\right)\xi_t.$$

- We have shown $\mathcal{I}(t), \mathcal{J}(t), \mathcal{K}(t)$ and $\mathcal{L}(t)$ for all $t \geqslant T_1$. Now we use them to prove $\mathcal{M}(t)$ for all $t \geqslant T_1$:

$$
\begin{aligned}
|h_{t+1}| &= \left(1 - \eta\left(\alpha_t^2 + \beta_t^2\right) + \eta^2\left(\alpha_t\beta_t h_t + \alpha_t^2\beta_{t,\perp}^2 + \beta_t^2\alpha_{t,\perp}^2 + \alpha_{t,\perp}^2\beta_{t,\perp}^2\right)\right)|h_t| + \eta\alpha_t\beta_t\xi_t - \eta^2\sigma_1\alpha_{t,\perp}^2\beta_{t,\perp}^2 \\
&\leqslant \left(1 - \frac{1}{2}\eta\left(\alpha_t^2 + \beta_t^2\right)\right)|h_t| + \eta\alpha_t\beta_t\xi_t \\
&\leqslant \left(1 - \frac{1}{2}\eta \cdot 2c_1\sigma_1\right)|h_t| + \eta\sigma_1\xi_t \\
&= (1 - c_1 c_{step})|h_t| + c_{step}\xi_t.
\end{aligned}
$$

Here we have used $\eta \leqslant \frac{\alpha_t^2 + \beta_t^2}{2\left|\alpha_t\beta_t h_t + \alpha_t^2\beta_{t,\perp}^2 + \beta_t^2\alpha_{t,\perp}^2 + \alpha_{t,\perp}^2\beta_{t,\perp}^2\right|}$, which is clearly true when $c_{step}$ is small enough.

Therefore, we have finished the proof of Lemma C.2. $\qquad\square$