[Reviews · NeurIPS 2018]

Reviewer 1



Edit after author feedback, I do not wish to change my score: - To me balanced positive quantities is not only about their difference. They should have similar order of magnitude, the difference between 1e-13 and 1 is pretty small but they are clearly unbalanced. - The author put a lot of emphasis on boundedness of iterates suggesting connections between their result and boundedness. - The authors do claim more than small difference between consecutive layer weight norm: lines 160 to 163. - I still think that writting down that a polynomial is not smooth in a math paper is not professional. - The author do abuse O notation: it involves a "forall" quantifier on a logical variable: en.wikipedia.org/wiki/Big_O_notation The same goes for "Theta" and "poly" notation. None of the statements of the paper involving these notations has this feature: the variables epsilon, d, d1 and d2 are fixed and are never quantified with a "forall" quantifier. The fact that a notation is standard does not mean that it cannot be misused. The authors consider deep learning models with a specific class of activation functions which ensures that the model remains homogeneous: multiplying the weight of a layer by a positive scalar and dividing the weights of another layer by the same amount does not change the prediction of the network. This property is satisfied for relu types activations. A consequence is that the empirical risk is not coercive, a common assumption in convergence analysis of first order methods. In order to circumvent this issue, the authors claim that gradient descent algorithm has a regularization effect, namely that it tends to ballance the magnitude of the weights of each layers such that they remain roughly of the same order. The main results are as follows: - The authors prove that difference of norm squared of consecutive losses is a constant over continuous (sub)-gradient trajectories for relu networks. - The authors use this intuition to provide results for matrix factorization without constraint: convergence to solution for decaying step size and convergence to a solution for constant step size in the rank 1 setting. Main comments: The abstract and introduction are very motivating and attractive. The idea of looking at invariances is nice and the balancing effect is a neat concept. Unfortunately, the content of the paper does not really reflect what the author claim in the first place: - The continuous time result is not really balancing anything and the authors clearly over interpret what they obtained. - 6 / 8 of the paper is dedicated to continuous subgradient trajectories in deep learning but 1 page of technical proof is dedicated to it (counting duplication) and 9 pages are dedicated to matrix factorization. While the arguments look correct to me, it looks like the main content of the paper is about gradient descent for unconstrained matrix factorization and the connection with deep learning, despite being a nice observation looks a bit like hand waving. Some elements of the mathematical presentation could be improved. Details: $$ About "auto-balancing" $$ The abstract says "This result implies that if the weights are initially small, gradient flow automatically balances the magnitudes of all layers" This is wrong. Nothing in Theorems 2.1 and 2.3 and Corollary 2.1 says that the weights will adjust and balance. Nothing even says that they should be bounded. The difference remains constant, but the weights could well diverge. The paragraph after corollary 2.1 affirms without any supporting arguments that the proposed result explains convergence. In particular, line 161-162 is wrong even in the simple case of matrix factorization. The fact that two numbers a and b satisfy a-b small does not mean that a or b are small. It seems to me that Theorem 2.1 is actually a special case of Theorem 2.3. Why giving a detailed proof for both of them then? $$ Why Gaussian random initialization? $$ Theorem 3.1 and 3.1 are stated with high probability over the random Gaussian initiailization. This is by no mean sufficient. Is 3/4 a high probability? Do the authors mean that it can be arbitrarily close to 1? How to tune the parameters then? Without a proper quantification, the result is of very little value. This can also be seen in the proof of Lemma 3.1, line 453, the proper way to present it would be to say which event is considered, to condition on that event and to estimate the probability of this event to be observed. Furthermore, I do not understand why the authors need Gaussian initialization. The main proof mechanism is that if initilization is close enough to the origin, then iterates remain bounded. This has nothing to do with Gaussianity. The authors could choose uniformly in a compact set, a hypercube is easy to sample from for example. The proof arguments would follow the same line but hold almost surely over the random initialization. The authors would avoid the non quantitative high probability statement and the incorect use of epsilon / poly(d). $$ Confusion about smoothness $$ There is a confusion throughout the paper about the notion of smoothness: - Line 35: f is not smooth. - Line 105: the objective is not smooth - Line 106: balancing implies smoothness - Line 162: the objective may have smoothness - Line 402: we prove that f is locally smooth This is completely misleading and mixes two important notions: - Smoothness is the property of being continuously differentiable, eventually several times, the more times, the more smooth (see wikipedia "smoothness" math page). - Optimization and ML people introduced notions of \beta-smoothness, which is just Lipschitz continuity of the gradient. The two notions are complementary. From an analysis point of view, being non smooth is the contrary of being smooth: not differentiable or extended valued for example. However being non-smooth is not equivalent to "not being beta-smooth" or more precisely "not being beta-smooth for any beta". To my understanding the appropriate notion of "not being beta-smooth for any beta" is that of stiffness (see for example Convex Analysis and Minimization Algorithms I: Fundamentals, Jean-Baptiste Hiriart-Urruty, ‎Claude Lemarechal page 375), that is smooth but with a modulus so big that it looks like nonsmooth numerically. This notion is pretty common in numerical analysis (see "Stiff equation" on wikipedia). The discussion in the main text is absolutely missleading: f is not smooth? f is a polynomial, it is infinitely smooth but it does not have a global smoothness modulus. Balancing implies smoothness? Smoothness is already here, twice continuous differentiability implies beta-smoothness on any bounded set. We prove local smoothness for f? As a polynomial, f is locally smooth, this does not require a proof, the authors actually compute a smoothness modulus. $$ Is the statement 5 = O(4) true or false? $$ In Theorem 3.1, the authors use N(0, \epsilon / poly(d)). Would 10 be $\espilon / poly(d)$ for example? Similarly in 266, $d_1 = \theta(d_2)$, would $1000$ be $\theta(500)$? or not? Asymptotic notations are asymptotic because they describe limiting behaviour which means stabilisation when some parameter increases or decreases. In the first statement, epsilon and d are fixed, similarly, d_1 and d_2 are fixed in the second statement. In addition to be mathematicaly meaningless, this provides no clue about how to choose variance in Theorem 3.1. $$ Minor remarks $$ - Notations the notation [.] is both used for the elements of a vector and for a set of integers and this sometimes makes reading not so easy. - Line 139, may be the authors could mention that the only homogeneous functions in one variable are more or less Relu like functions (linear when restricted to R+ and R-). - Equation (5), why should a solution of the differential inclusion exist, why this problem is well posed? - Line 197, it would be nice to have a more precise reference to a particular result and explain why this chain rule should hold for the specific situation considered in this work. - Equation after line 206, an equality is missing. - More explaination would be welcome after line 207. - Line 291, connections between optimization and ODE have been known for a while and investigated extensively in the early 2000s by the mathematical optimization community, in particular Hedy Attouch and his team. - Line 381, typo, theorem 2.3 - Line 435, argument is missing for boundedness of V - Equation (16), there is a typo. - I did not have time to check the rank 1 case proof in the appendix.

Reviewer 2



The paper studies the dynamic effects of the continuous gradient flow upon the scaling of the weights in multilayer Neural Networks. IMHO, the main result of the paper is Theorem 2.1 (line 148) that establishes the existance of (N-1) integral of motions where N is the number of layers. It follows directly from eq. (6) [line 149] that the L2-norm of all the layers are are at the constant offsets from each other and that it is sufficient to follow the dynamics of L2 norm of the 1st layer to recover all the other norms provided the initial offsets are known. This is an interesting result and if true it warrant further studies. However the paper is incomplete, for it failed to consider several very important cases: (1) Run a simple experiment with a Feed Forward Neural Network (e.g. 3-layer MLP) and observe the dynamics of L2-norms of Weights at each layer to confirm or disprove eq (6). (2) Effects of discrete time flow on validity of Theorem 2.1 are not studied. (3) Prove of Theorem 2.1 is given based on the gradient flow (5) [line 143]. Will it hold for SGD with Momentum? (4) Batch-normalization introduces extra scaling in X-space that can effect the relationships used in derivation of Theorem 2.1.

Reviewer 3



Summarizing the main ideas: This paper investigates the performance of gradient descent as a first order optimization method in multi-layer homogeneous functions. The main idea has been presented in two consecutive sections. First section is on theoretical investigation of the implicit regularization achieved by the gradient descent with infinitesimal step size in neural networks. The main contribution in this part is proving a theorem that shows in homogeneous models, gradient flow can balance the incoming and outgoing weights for each neuron. So, by this proof, a theoretical basis is established that shows weights across layers are balanced. After providing the invariance property for linear activation functions, similar analysis was developed for sparse neural networks like convolution ones. Then, the authors investigated the gradient descent with positive step size. The authors were able to prove global convergence and balancing property from random initialization. Previous analysis helped authors to prove global linear rate of convergence to a global minimum in the matrix factorization problem. Strength and weakness (quality, clarity, originality, and significance): It is the first paper analyzing difficulties one faces in optimizing multi-layer homogeneous functions. If the gradient changes smoothly, then it is possible to prove global converges as discussed in related work of the paper. The main difficulty here lies in the fact that the gradient is not changing smoothly. To my knowledge, existing global convergence results that I am aware of does not work for multi-layer homogeneous functions. Therefore, I consider this paper as an important contribution that provides convergence analysis for a class of non-smooth non-convex unconstrained optimization problem that is optimizing multi-layer homogeneous functions. I think this paper also elucidates the capabilities of batch normalization commonly used in deep learning. It is thought that batch normalization helps in regularizing and improving gradient flow in addition to reducing number of local minima. This paper shows that without batch normalization, GD already has regularization and good gradient flow. Therefore, the main power of batch normalization could be the power of avoiding local minima. In spite of the page limit, the details and necessary information were clearly delivered in this paper. The overall concept of the paper is clear and it is easy to read paper. One of the problems about this paper is the lack of enough experiments supporting the proofs of the paper. The simulation is limited to a matrix factorization experiment of Fig. 1. Similar investigation for deep neural network with homogenous activation function should be considered.

Reviewer 4



Edit: I confirm a weak accept. The figures presented in the author's feedback and their interpretation are not sufficient. We clearly see that there is a difference between the prediction of corollary 2.1 and the given results in the first 100 epochs. This is probably due to discretization, but there is no experimental result about the effect of the choice of the step size. Do we tend to have trajectories of the norms of the weights matrices verifying corollary 2.1 when the step size tends to zero? If so, we should be able to characterize such convergence (speed...). If not, why? However, the article raises many interesting questions, and provides an analysis of a side-effect of the SGD, which is not so commo. The paper proposes an explanation for the stability of the SGD algorithm in neural networks with homogeneous activation functions. The main result is the proof that the difference between the norms of the weight matrices of two consecutive layers remains constant (with a infinitely small gradient step). The secondary result concerns the stability of the gradient descent for matrix factorization. The article is part of the field of theoretical analysis of dynamics of neural networks. The main contribution is the study of a side-effect of the SGD, which facilitates the convergence. The main result is well explained and the proof is clear. The explanation of such indirect regularization through SGD should be very helpful to understand the efficiency of neural networks, notably to identify the contribution of the optimizer. Nevertheless, there is (almost) no experimental validation of the theoretic prediction, especially in the case of the main result (case of neural networks). Moreover, the only plot giving the ratio between the norms of two matrices (figure 1b) contradicts theorem 3.1 for the first 300 epochs. Unfortunately, there is no discussion about it. In general, the paper is well written, but the purpose of the article remains unclear. Is it an article about matrix factorization or about neural networks ? The main result seems to be corollary 2.2 (about neural networks), but the only figures given are about matrix factorization. Some improvement could be made to clarify the intention. Overall, the main result is very interesting, and should be helpful to understand the consequences of the use of SGD as optimizer. But there is no experiment to confirm it, even in some particular and simple cases. The study of matrix factorization seems to be irrelevant in this kind of article.